# Prior flavivirus immunity skews the yellow fever vaccine response to cross-reactive antibodies with potential to enhance dengue virus infection

The yellow fever 17D vaccine (YF17D) is highly effective but is frequently administered to individuals with pre-existing cross-reactive immunity, potentially impacting their immune responses. Here, we investigate the impact of pre-existing flavivirus immunity induced by the tick-borne encephalitis virus (TBEV) vaccine on the response to YF17D vaccination in 250 individuals up to 28 days post-vaccination (pv) and 22 individuals sampled one-year pv. Our findings indicate that previous TBEV vaccination does not affect the early IgM-driven neutralizing response to YF17D. However, pre-vaccination sera enhance YF17D virus infection in vitro via antibody-dependent enhancement (ADE). Following YF17D vaccination, TBEV-pre-vaccinated individuals develop high amounts of cross-reactive IgG antibodies with poor neutralizing capacity. In contrast, TBEV-unvaccinated individuals elicit a non-cross-reacting neutralizing response. Using YF17D envelope protein mutants displaying different epitopes, we identify quaternary dimeric epitopes as the primary target of neutralizing antibodies. Additionally, TBEV-pre-vaccination skews the IgG response towards the pan-flavivirus fusion loop epitope (FLE), capable of mediating ADE of dengue and Zika virus infections in vitro. Together, we propose that YF17D vaccination conceals the FLE in individuals without prior flavivirus exposure but favors a cross-reactive IgG response in TBEV-pre-vaccinated recipients directed to the FLE with potential to enhance dengue virus infection.

Flaviviruses are distributed globally and are rapidly spreading due to international trade and travel, poorly planned urbanization, and ecological and climate changes[1,2]. Human pathogenic flaviviruses comprise over thirty antigenically related viruses[3]. Tick-borne encephalitis virus (TBEV), mosquito-borne yellow fever (YFV), dengue (DENV), Zika (ZIKV), West Nile (WNV), and Japanese encephalitis virus (JEV) are flaviviruses with the potential to cause severe disease, representing a leading cause of morbidity and mortality worldwide and, in the case of DENV, infecting up to 400 million people annually[4]. Their global distribution, high prevalence, and increasing vaccination coverage have resulted in a rising number of individuals with immune experience to flaviviruses. Hence, cross-reactive immunity at the time of vaccination or natural infection with another member of the *Flaviviridae* family is likely to occur.

Flaviviruses share a similar structure, mode of cell entry, and mechanisms of maturation and assembly. They are spherical enveloped viruses of about 50 nm in diameter, containing a single-stranded, positive-sense RNA genome of about 11,000 nucleotides encoding for a

✉ e-mail: giovanna.barba-spaeth@pasteur.fr; simon.rothenfusser@med.uni-muenchen.de

polyprotein that is post-translationally cleaved into three structural proteins: capsid, pre-membrane (prM) and envelope (E), and seven non-structural proteins. In mature virions, the structural proteins are inserted into the host-derived lipid bilayer in an icosahedral architecture where 180 units of the E protein cover the surface of the virion in 90 head-to-tail homodimers. The E protein is the main target of the neutralizing antibody response and its structure and dynamics define the epitope landscape of the virus[5,6]. The E protein consists of three structurally defined domains (DI, DII and DIII), connected to two transmembrane domains by three stem helices. DII contains the highly conserved hydrophobic fusion loop (FL)[7]. A significant portion of the antibody response is directed towards the fusion loop epitope (FLE), which is generally concealed in the dimeric arrangement of the E protein. Although FLE is immunodominant and cross-reactive, its poor accessibility renders fusion loop antibodies weakly neutralizing[7–9]. For effective anti-flavivirus immunity, a relevant fraction of the humoral response targets complex quaternary epitopes spanning regions in different E units. In DENV and ZIKV, the E dimer epitope (EDE), encompasses regions in DI and DIII of one protomer and DII of the opposing protomer requiring a dimer subunit to be exposed. Upon binding, EDE antibodies crosslink the dimer, preventing the conformational changes necessary for fusion. EDE antibodies have been precisely mapped for DENV and ZIKV, but dimeric quaternary epitopes exist for other flaviviruses[7,10–12].

Notably, antibody cross-reactivity has been described even among distantly related flaviviruses, which can impact the immune response and clinical course in secondary infections[7,13]. Indeed, cross-reactive antibodies can facilitate virus entry via Fcγ-receptor-mediated phagocytosis in a process known as antibody-dependent enhancement (ADE) of infection[14]. As a consequence, ADE of DENV infection is clinically associated with an enhanced risk of severe dengue disease in secondary DENV exposures in humans[15–18]. Likewise, pre-acquired cross-reactive immunity can impact live vaccine responses in different ways. If pre-existing immunity propitiates virus neutralization, faster clearance, or epitope masking it might lead to suboptimal boosting of immunity[19]. Alternatively, cross-reactive antibodies can enhance productive infection of antigen-presenting cells via ADE, leading to an increased immune response[20,21]. Lastly, immune imprinting from an earlier immunization may hamper the ability to mount an adequate response against a new antigenic challenge[22].

The YF17D vaccine induces life-long protective immunity and is considered one of the most effective vaccines ever developed[23,24]. Prior studies have highlighted dimer-dependent epitopes and the FL-proximal region of the E protein as important neutralizing sites for YFV[25–28]. In sera of human vaccinees, the response predominantly targets DII, with only a minor fraction targeting DIII[29,30]. Inter- or intra-dimer epitopes, equivalent to EDE, have not yet been structurally mapped for YFV[7], and studies in humans have been restricted to E monomer moieties which account for only a small part of the neutralizing response[30]. Given the high prevalence of flaviviruses, YF17D is frequently administered to individuals with flavivirus immune experience. Although the effects of sequential exposures to different flavivirus infections or vaccinations on the immune response have been previously investigated[20,31–34], their impact on YF17D-induced antibody specificities and vaccine immunogenicity remains a relevant and still open question.

In this study, we investigated how pre-existing immunity to TBEV, an antigenically related but phylogenetically distant flavivirus, affects the immunogenicity and epitope immunodominance of the YF17D vaccine. Using a longitudinal cohort of 250 participants and a separate cohort of 22 donors sampled one year after vaccination, we found that YF17D elicits non-cross-reactive but efficiently neutralizing antibodies in flavivirus-naïve individuals. In contrast, the antibody response in TBEV-experienced individuals is skewed towards the cross-reactive FLE. We further demonstrate in vitro that this response results in ADE of DENV and ZIKV infection.

## Results

### YF17D vaccination induces similar neutralizing antibody titers but expands a poor-neutralizing IgG response in TBEV-pre-vaccinated individuals

To investigate the effects of pre-existing cross-reactive immunity on the response to the live YF17D vaccine, we examined a longitudinal cohort of 250 YF17D healthy young vaccinees grouped based on their previous immunization with the inactivated TBEV vaccine (cohort-1). Given that TBEV vaccination is recommended in the region where this study was conducted, a representative fraction of participants self-reported a history of TBEV vaccination prior (>4 weeks) to study inclusion (n = 162; 64.8%). TBEV pre-immunity was verified through positive results for TBEV neutralization by plaque reduction neutralization assay (PRNT) and positive detection of anti-TBEV-DIII IgG antibodies using enzyme-linked immunosorbent assay (ELISA). Individuals with discrepancies between self-reported vaccination status and serological assays were excluded from the analysis. The final study cohort-1 comprised 139 participants pre-vaccinated against TBEV and 56 TBEV-unvaccinated individuals. Different sexes, ages, and BMI values are equally distributed in both subgroups (Fig. 1; Table S1). The exact strain, number of doses, and timing of TBEV vaccinations had not been documented. Consequently, TBEV-experienced donors showed heterogeneity in their TBEV neutralizing titers prior to YF17D vaccination (Fig. 2B).

The neutralizing antibody titer against YF17D, commonly used as a correlate of vaccine-induced protection, was equally strong at day 28 post-vaccination (pv) in both flavivirus-naïve and experienced individuals. This result indicates that TBEV-pre-immunity does not impair the neutralizing antibody response to YF17D (Fig. 2A). Conversely, YF17D immunization did not alter the neutralizing activity against TBEV, indicating that the YF17D vaccine did not generate cross-neutralizing antibodies to TBEV (Fig. 2B).

The presence of YF17D virion-specific IgM in serum was measured longitudinally by ELISA. The IgM titer reached a plateau between day 14 and 28 pv and was comparable in both groups of vaccinees as confirmed in IgG-depleted serum samples (Fig. 2C, D). Unlike the IgM response, participants with a prior TBEV exposure had YF17D cross-reactive IgG antibodies already at baseline and, upon vaccination, the IgG titer was further boosted resulting in a 100-fold higher titer compared to the TBEV-unvaccinated group. The same dynamic was observed for E protein (Fig. 2E) and full-virion specific IgG (Fig. S1B, C). The IgG titer continued to increase from day 14 to day 28 pv, at which timepoint all the study participants had seroconverted. While TBEV-pre-vaccinated donors showed an anti-E IgG response already at day 14 pv, only a fraction of the TBEV-unvaccinated participants generated detectable anti-E IgG levels at that timepoint, suggesting an earlier response to YF17D in individuals with immune experience (Fig. 2E).

To assess the generation of vaccine-specific B cells, we implemented a soluble (s)E-specific ELISpot assay. The number of plasmablasts was quantified directly ex vivo and the total number of sE-specific B cells was measured following the differentiation of B cells into antibody-secreting cells. sE-specific IgG secreting plasmablasts peaked at day 14 pv in TBEV-pre-vaccinated participants but were below detection for unvaccinated donors (Fig. 2F). The total amount of sE-specific B cells was in line with the IgG levels measured in serum with a significantly higher B cell number in flavivirus-experienced individuals. Likewise, sE-specific B cells were detected earlier, on day 14, in TBEV-pre-immunized individuals (Fig. 2G).

Collectively, these results indicate that TBEV pre-immunization does not hinder the response to YF17D. TBEV-pre-immunized individuals have cross-reactive IgG antibodies to YF17D and experience an earlier and stronger IgG response.

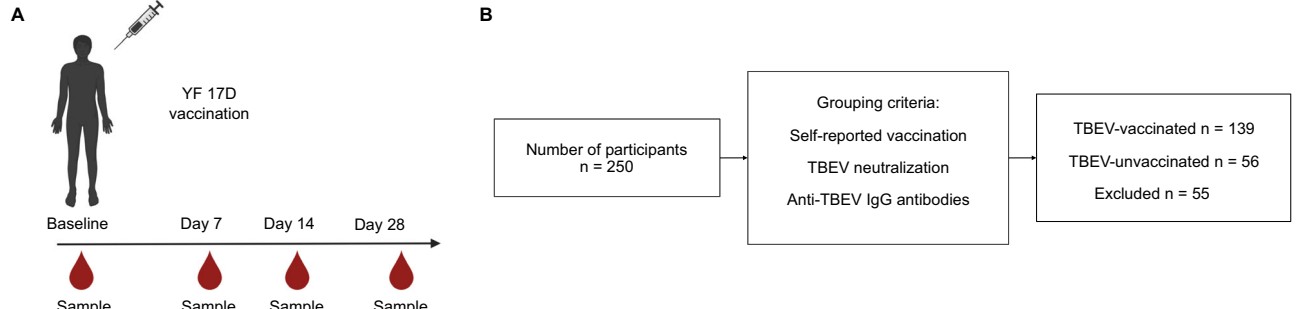

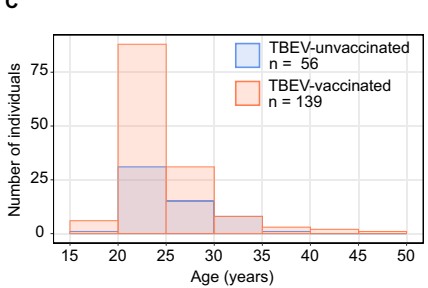

| | All | TBEV Vaccinated | TBEV Unvaccinated |
|---|---|---|---|
| TOTAL | 250 | 139 | 56 |
| Sex (%) | | | |
| female | 169 (67.6) | 99 (71.2) | 38 (67.9) |
| male | 81 (32.4) | 40 (28.8) | 18 (32.1) |
| Age median (range) | 24 (19-47) | 24 (20-46) | 24.5 (21-38) |
| BMI median | 22.7 | 22.25 | 21.80 |
| above 25 (%) | 30 (12) | 12 (8.6) | 4 (7.1) |
| below 18.5 (%) | 8 (3.2) | 3 (2.2) | 4 (7.1) |

**Fig. 1 | Cohort-1. Overview of study participants. A** Diagram representing the longitudinal PBMC, serum, and plasma sample collection of 250 participants. Samples were collected at baseline (immediately before vaccination) and on days 7, 14 and 28 post vaccination. Prepared with Biorender (www.biorender.com) **B** Flow chart of cohort members grouping according to TBEV pre-vaccination status. 139 individuals self-reported having received at least one TBEV-vaccine dose and contained neutralizing antibodies and anti-TBEV IgG at baseline. 56 TBEV-unvaccinated individuals were classified based on a self-reported negative vaccination history validated by the absence of detectable anti-TBEV IgG and neutralizing capacity at baseline **C** Histogram depicting the age distribution of the 139 TBEV-vaccinated participants (in orange) and the 56 TBEV-unvaccinated donors (in blue). **D** Table summarizing cohort-1 characteristics for all 250 individuals and separated according to TBEV-pre-vaccination status.

## IgM antibodies account for most of the neutralizing capacity at days 14 and 28 post-immunization

The observed differences in the IgG titer did not correspond with the similar neutralization capacity between both groups of vaccinees. When groups were analyzed separately, the IgG titer at day 28 correlated weakly with neutralization ($R = 0.29$, p 0.04 and $R = 0.34$, $p < 0.0001$), whereas there was a stronger association with the IgM levels ($R = 0.55$, $p < 0.0001$ and $R = 0.64$, $p < 0.0001$ for TBEV-unvaccinated and experienced individuals, respectively) (Fig. 2H, I). We depleted serum of the IgM or IgG fractions in a subgroup of the study cohort-1 to precisely define the contribution of IgG and IgM antibodies to neutralization. We confirmed that depletions were successful, without loss of the alternative antibody fraction (Fig. S1D). We observed that the IgM fraction was the main mediator of virus neutralization, accounting for approximately 75% of the neutralizing titer on day 28 (Fig. 2J, L). Consistently, IgG depletion led only to a 25% loss of neutralization capacity on days 14 and 28 (Fig. 2J, K, and S1D, E). IgM-depleted sera of TBEV-pre-immunized individuals showed a significantly higher neutralizing capacity, indicating that the increased IgG titer contributed partly to virus neutralization (Fig. 2L). There was no significant difference in the IgM antibody titers and IgM-mediated neutralization between groups (Fig. 2D, J, K).

Thus, on day 28 the IgM fraction is primarily responsible for the neutralization capacity and is equally strong regardless of the pre-vaccination status. The IgG fraction can mediate neutralization, but the boosted response in TBEV-experienced individuals is predominantly directed towards poorly-neutralizing epitopes.

## TBEV-induced immunity mediates ADE of YF17D virus infection

As reported by Chan et al. 2016, cross-reactive non-neutralizing or sub-neutralizing IgG antibodies can mediate FcγR engagement and increase vaccine immunogenicity via ADE of YF17D virus infection[20]. Given that TBEV-pre-vaccinated individuals had YF17D cross-reactive antibodies at baseline, we measured whether they could facilitate YF17D infection of FcγR expressing cell lines (THP-1 and K562). We observed an enhanced infection of a Venus-fluorescent YF17D virus in both cell lines in the presence of serum from TBEV-pre-immunized donors. ADE was IgG-dependent, as it was absent in IgG-depleted serum but not in IgM-depleted serum, and it was inhibited in the presence of FcγR-blocking antibodies (Fig. 3A and S2A, B, C). YF17D infection enhancement was observed exclusively with sera from TBEV-experienced individuals (Fig. 3B) which showed the highest enhancing titer, calculated as area under the curve (AUC) (Fig. 3C).

In addition, we quantified the IgG fraction targeting DIII. DIII is often used for serological diagnosis[35] and although cross-reactive epitopes have been described[36,37], responses to DIII are generally virus-specific[38]. As observed for anti-sE IgG antibodies, TBEV-pre-vaccinated individuals had a stronger IgG response against YF17D-DIII (Fig. 3D). We then compared the fold change in anti-YF17D-DIII and anti-TBEV-DIII IgG titers between baseline and day 28 pv. Interestingly, the strong expansion of anti-YF17D-DIII-reactive IgG (10-100-fold) contrasted with the moderate increase in TBEV-DIII reactive IgG (<10-fold) (Fig. 3E−G). Since the IgG fraction targeting TBEV-DIII was not expanded to the same extent, we conclude that YF17D is not only boosting cross-reactive TBEV-induced memory responses but also triggers antibodies towards previously unseen, non-cross-reactive epitopes in DIII, potentially via enhanced immunogenicity.

The correlation between the baseline levels of sE-specific IgG and the baseline enhancing titer with the post-immunization IgG response to sE and DIII, as well as the post-vaccination neutralizing titers, suggests that ADE of YF17D infection could mediate an increase in vaccine immunogenicity (Fig. 3H, S2D, E, F). To gain insight into the size effects

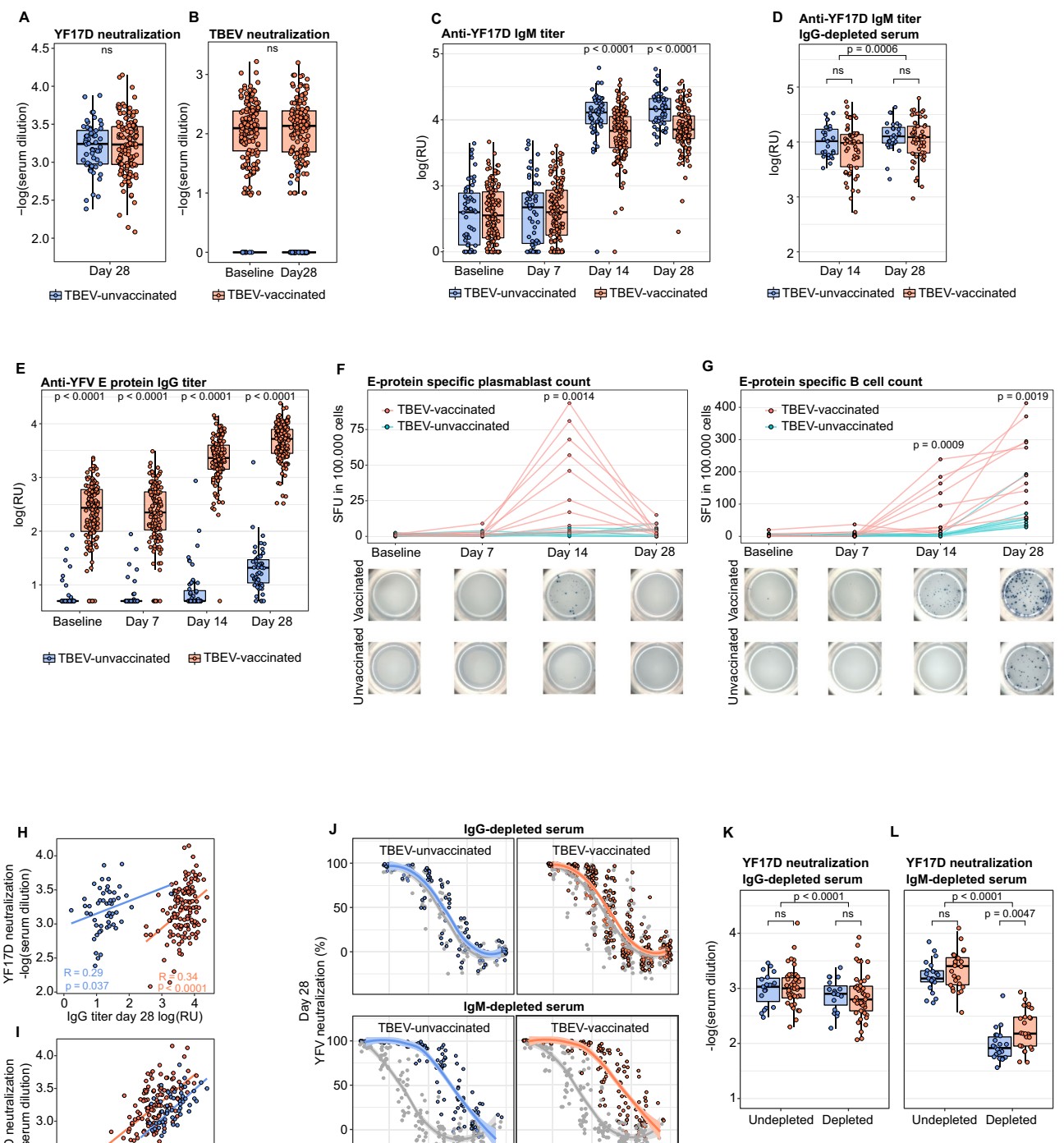

of ADE of YF17D infection on vaccine immunogenicity, we grouped the vaccinees into quartiles based on their enhancing and baseline IgG titers. This approach removed individuals with intermediate TBEV-vaccine-induced IgG levels from the analysis and therefore improved the precise identification of true differences between the high and low vaccine enhancers. Participants in the highest quartile of the enhancing titer had significantly higher neutralizing titer and IgG levels against sE and DIII (Fig. 3I) than the participants in the lowest quartile. The same associations were observed with baseline IgG quartiles (Fig. 3J). Thus, within the TBEV-pre-vaccinated group, ADE of YF17D infection was associated with increased vaccine immunogenicity.

Altogether, in addition to an anamnestic response, these results suggest that TBEV-pre-immunity may increase the magnitude and breadth of the humoral response to YF17D vaccine facilitated by ADE of YF17D infection.

**YF17D effectively primes for non-cross-reactive antibodies in flavivirus-naïve vaccinees but expands a pan-flavivirus cross-reactive IgG response in TBEV-experienced individuals**

The IgG response in both groups was investigated for cross-reactivity with other members of the *Flaviviridae* family. Using an indirect immunofluorescence test, the IgG and IgM binding to ZIKV, JEV, WNV,

**Fig. 2 | Comparison of the YF17D-induced antibody and B cell responses between TBEV-vaccinated and unvaccinated participants. A** YF17D 80% neutralization titer at day 28 post-vaccination for TBEV-vaccinated ($n = 137$) and unvaccinated ($n = 56$) individuals. **B** TBEV 80% neutralization titer before and 28 days after YF17D vaccination for TBEV-vaccinated individuals ($n = 137$) and TBEV-unvaccinated individuals ($n = 56$). **C** Longitudinal YF17D virion-specific IgM response in TBEV pre-vaccinated (baseline, $n = 132$; day 7, n = 128; day 14, n = 128; day 28, $n = 129$) and TBEV unvaccinated donors (baseline, $n = 54$; day 7, $n = 54$; day 14, $n = 54$; day 28, $n = 53$). **D** YF17D virion-specific IgM titer on days 14 and 28 in IgG-depleted serum for TBEV pre-vaccinated ($n = 56$) and TBEV unvaccinated donors ($n = 28$). **E** Longitudinal YF17D anti-E protein-specific IgG titers for TBEV-vaccinated (baseline, $n = 136$; day 7, n = 131; day 14, n = 132; day 28, n = 133) and TBEV-unvaccinated donors (baseline, $n = 52$; day 7, $n = 52$; day 14, $n = 52$; day 28, $n = 52$). **F, G** Plasmablasts and total sE-specific longitudinal B-cell response quantified by ELISpot and depicted as spot-forming units (SFU) in 100,000 PBMC ($n = 10$ TBEV-vaccinated and $n = 9$ TBEV-unvaccinated donors). Spot pictures are shown for a representative example of a TBEV-pre-vaccinated and unvaccinated individual. Significance compares B cell counts between groups on days 14 (**F**) and 28 (**G**).

**H, I** Spearman correlation between the YF17D polyclonal neutralizing titer of sera on day 28 with the IgG titer (TBEV-vaccinated $n = 133$ and TBEV-unvaccinated donors $n = 52$) and IgM titer (TBEV-vaccinated $n = 129$, TBEV-unvaccinated donors $n = 53$). **J** Neutralization curves of undepleted polyclonal serum and IgG or IgM-depleted serum (in grey) for TBEV-pre-vaccinated ($n = 45$ and 26, respectively) and unvaccinated individuals ($n = 19$ and $n = 22$, respectively). Quantification of the 80% neutralization cutoff before and after IgG (**K**) and IgM (**L**) depletions shown in **J**. TBEV-vaccinated participants are depicted in orange and TBEV-unvaccinated in blue. Boxplots show a horizontal line indicating the median and lower and upper hinges corresponding to the first and third quartiles. The lower and upper whiskers extend to 1.5x IQR (interquartile range) from the respective hinge. The curve fitting in **J** was done with local regression with a 0.95 confidence interval. Statistical significance between TBEV-vaccinated and unvaccinated individuals is shown above every comparison and was estimated with a two-sided Mann-Whitney test. Statistical significance between undepleted and depleted samples (**K** and **I**) and between timepoints (**D**) was calculated with a two-sided Wilcoxon signed-rank test. $P$ values above 0.05 are considered non-significant (ns).

TBEV, YFV, and all serotypes of DENV was measured in a cohort subset (Fig. 4, S3). At baseline, TBEV pre-vaccination induced an IgG response cross-reactive with all flaviviruses at similar magnitude. This pan-flavivirus cross-reactive IgG signature was boosted upon vaccination with the YF17D vaccine. In contrast, the YF17D vaccine induced an IgG response that targeted uniquely YFV in unvaccinated individuals (Fig. 4A). Consistent with previous studies, the IgM signature was YFV-specific and could not be detected at baseline (Fig. 4B).

These results highlight the capacity of the vaccine strain of YFV to prevent a cross-reactive response when administered to flavivirus-naïve individuals.

## Pre-existing immunity changes YF17D immunodominance and misdirects the response to FLE

Depending on previous flavivirus exposure, YF17D induces a differential cross-reactivity pattern while eliciting a comparable neutralizing response. We hypothesized that the neutralizing capacity is predominantly driven by antibodies targeting quaternary dimeric epitopes, equivalent to EDE (EDE-like), and the FL-proximal region, whereas cross-reactive antibodies target the immunodominant FLE. To better understand this change in the immunodominance, we designed a set of recombinant sE protein mutants for the study of the IgG response to different epitopes (Fig. 5A). Besides the monomeric sE protein containing all three ectodomains and variants consisting of either only DI-II or only DIII, we designed constructs displaying quaternary dimeric epitopes to reproduce the epitope landscape of YF17D more realistically. The substitution S253C in DII allows the formation of an inter-protomer disulfide bond across the two sE protomers, generating a covalently bound dimer[39]. This construct (hereinafter referred to as breathing-dimer) retains the ability to oscillate and exposes EDE-like dimeric epitopes, FLE, and the FL-proximal region. Furthermore, a W101H substitution was introduced in the breathing-dimer setting to disrupt the FLE (breathing-dimer[W101H]) (Fig. 5A). In addition, we produced a locked-dimer E protein by introducing a double substitution L107C and T311C following the strategy used by Rouvinski et al (2017) and Slon-Campos et al (2019) with DENV and ZIKV. This construct displays quaternary epitopes on a pre-fusion dimeric structure that is bridged with two disulfide bonds between DI and DIII of opposing protomer units (Fig. S4A, B). Correct folding and epitope exposure of the recombinant proteins were verified by binding with specific antibodies using ELISA and SEC analysis (Fig. S6). Both the FLE KO construct (breathing-dimer[W101H]) and the locked dimer failed to bind antibodies recognizing the FLE but were recognized by an antibody binding DII outside of the FL. Both sE monomer and the breathing-dimer were recognized by fusion loop specific antibodies (Fig. S6).

The comparison between the breathing-dimer and breathing-dimer[W101H]-specific IgG titers serves to measure the fraction targeting the FLE. For TBEV-experienced individuals, the antibody fraction targeting the breathing-dimer was significantly reduced in baseline samples and at day 28 by the W101H mutation (45 and 64% reduction respectively) whereas TBEV-naïve individuals showed no significant difference (Fig. 5B). Additionally, the sE-specific IgG titer was quantified in binding competition assays with 4G2 (pan-flavivirus FLE-specific mAb) and 2D12 (YFV-neutralizing, non-cross-reactive anti-E mAb) (Fig. 5C, S5A). As anticipated, TBEV-pre-exposed vaccinees lost over 80% of the sE-IgG binding fraction at day 28 in competition with 4G2, while flavivirus-naïve individuals ranged from 0 to 60% binding loss, demonstrating that FLE is a dominant binding site for the antibody response in TBEV-experienced individuals, but not in TBEV-unvaccinated individuals. Likewise, baseline antibodies also competed with 4G2 for binding (Fig. 5C). Consistently, binding loss caused by the W101H mutation and competition with 4G2 correlated with each other ($R = 0.65$, $p = 0.0012$), serving as cross-validation of these assays to quantify FLE antibodies in serum (Fig. 5D).

Additionally, we performed an ELISpot assay to quantify the number of epitope-specific circulating B cells. We observed that the number of breathing-dimer-specific B cells was larger for TBEV-pre-immunized compared to TBEV-unvaccinated participants. Approximately 50% of the specific-B cells in TBEV-pre-immunized donors (100 cells/100.000 lymphocytes) required the unmutated FL for binding (Fig. 5E). Consistently with serum antibody levels, TBEV-unvaccinated individuals had lower numbers of breathing-dimer specific B cells, although a relevant fraction produced antibodies requiring FL for binding. These B cells secrete antibodies that may be binding dimeric structures or FL-proximal regions whose binding site includes amino acids located in the FL (Fig. 5A, E). The locked-dimer-specific IgG and B cell response was in line with the findings showing increased responses in TBEV-experienced individuals (Fig. S4C, D).

Taken together, these results show that the IgG fraction targeting the FLE is dominant in TBEV-pre-immunized but not in TBEV-unvaccinated participants. However, the fusion loop region is a binding site for antibodies elicited in both groups.

## Antibodies targeting quaternary E-protein epitopes are central for neutralization of YF17D in TBEV-experienced and naïve vaccinees

To assess the neutralizing capacity of antibodies with different specificities, we performed antigen-specific IgG depletion from serum samples that had been pre-depleted of IgM antibodies (Methods and Fig. 5F, G). Depleting the IgM fraction allowed for a more precise

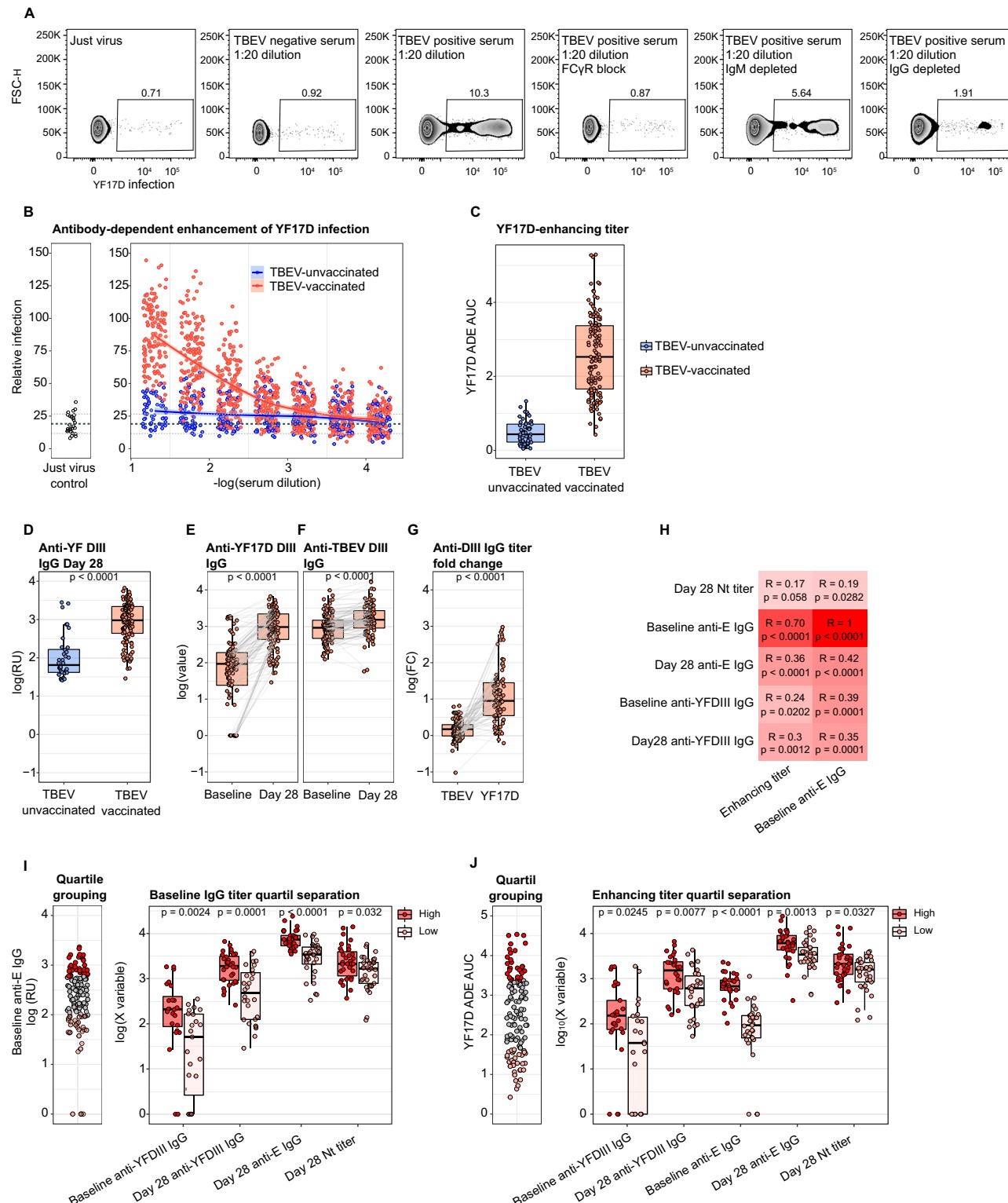

dissection of the main neutralizing sites targeted by the long-term, durable, IgG response. As expected, IgM removal greatly reduced the neutralizing capacity of the sera (Fig. 2G). Further depletion with the breathing-dimer protein resulted in a remarkable loss of neutralizing capacity in both TBEV-experienced and naïve individuals. The neutralizing titers of sera depleted with the breathing-dimer[W101H] antigen remained high, demonstrating that neutralizing epitopes include the FL as a binding site (Fig. 5G, left panels). In fact, the main chain of the FL is part of the EDE epitope in DENV[10]. Depletions performed with the

locked-dimer construct resulted in only a slight decrease in neutralizing capacity (Fig. S4E). Despite the occlusion of the FL in the locked-dimer, we thought this construct would deplete antibodies with dimeric specificities and would reduce greatly the serum neutralization activity. However, the direct modification of the FLE by the L107C mutation of this construct also had an impact on the integrity of the dimer epitope, affecting the ability to deplete antibodies targeting dimeric specificities. A comparable construct for dengue[40], although able to bind most of the dengue EDE antibodies, showed a reduction in

**Fig. 3 | Antibody dependent enhancement of YF17D virus infection mediated by TBEV vaccine-induced IgG. A** Flow cytometric determination of venus-YF17D virus infection of THP-1 cells in the absence or presence of cross-reactive serum from a TBEV-vaccinated individual. The conditions tested include polyclonal serum alone or in combination with FcγR-blocking antibodies and IgM or IgG-depleted serum. **B** ADE of YF17D mediated by study participants' serum (*n* = 132 TBEV-vaccinated, *n* = 53 TBEV-unvaccinated individuals). Virus infection was quantified in combination with serially diluted serum and was normalized against the enhancement of a 1:20 diluted enhancing-serum control carried for all measurements. Thick dashed line indicates the mean of the different just-virus controls and dotted lines define +/− 1 SD. The curve was fitted with local regression with a 0.95 confidence interval. **C** Quantification of the enhancing titer as AUC of the normalized virus infection across serial dilutions shown in **B**. **D** Comparison of anti-YF17D-DIII specific IgG titer at day 28 post-vaccination for both TBEV groups (*n* = 36 TBEV-unvaccinated, *n* = 117 TBEV-vaccinated individuals). Longitudinal quantification of anti-YF17D-DIII (**E**, *n* = 97 donors) and anti-TBEV-DIII (**F**, *n* = 114 donors) specific IgG in TBEV-experienced individuals. **G** Paired comparison of the DIII-specific IgG fold-change between day 28 and day 0 for TBEV and YF17D (*n* = 76 pairs). **H** Spearman correlation in TBEV-experienced individuals of baseline anti-E IgG and enhancing titers

versus YF17D vaccine-induced neutralizing antibody titer at day 28, anti-sE and anti-DIII IgG titers and baseline anti-YF17D-E and anti-YF17D-DIII IgG titers. Color intensity reflects the Spearman correlation coefficient. **I** Comparison of the first (*n* = 34) and fourth (*n* = 33) quartile group of YF17D-sE specific IgG at baseline and the anti-YFDIII IgG on baseline and day 28, the anti-YF17D-sE IgG titer on day 28 and the anti-YF17D neutralizing antibody titer on day 28 (high vs. low quartile *n* = 24/23, 32/28, 33/32, 34/33 respectively). **J** Comparison of the first (*n* = 33) and fourth (*n* = 33) quartile group of the Enhancing titers at baseline and the anti-YFDIII IgG on baseline and day 28, the anti-YF17D-sE IgG titer at baseline and day 28 and the anti-YF17D neutralizing antibody titer on day 28 (high vs low quartile *n* = 25/22, 28/30, 32/33, 32/33, 33/33, respectively). TBEV-vaccinated participants are depicted in orange and TBEV-unvaccinated in blue. Boxplots show a horizontal line indicating the median and lower and upper hinges corresponding to the first and third quartiles. The lower and upper whiskers extend to 1.5x IQR from the respective hinge. Statistical significance between TBEV-vaccinated and unvaccinated individuals (**D**) and between high and low quartiles (**I**, **J**) is shown above every comparison and was estimated with a two-sided Mann-Whitney test. Statistical significance in **E**–**G** was calculated with a two-sided Wilcoxon signed-rank test. P values above 0.05 are considered non-significant (ns).

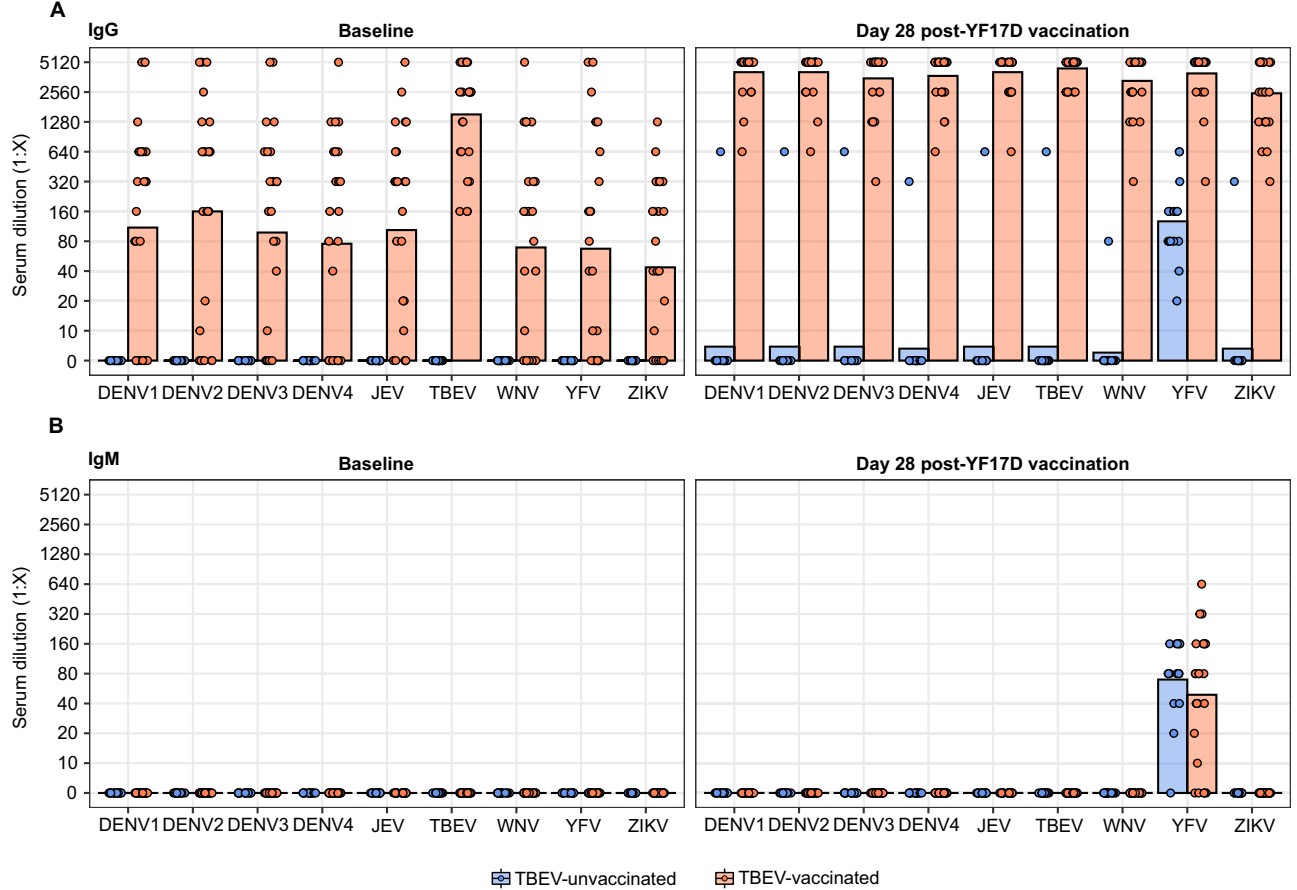

**Fig. 4 | IgG and IgM cross-reactivity signature before and after YF17D vaccination in TBEV-unvaccinated and pre-vaccinated individuals.** IgG (**A**) and IgM (**B**) subtype cross-reactivity evaluation before and after YF17D vaccination using an indirect immunofluorescent test for a panel of nine human-pathogenic flaviviruses: DENV 1–4, ZIKV, WNV, JEV, YFV and TBEV. A subgroup of *n* = 39 individuals was

tested (Fig. S3), out of which *n* = 15 were TBEV-unvaccinated and *n* = 24 TBEV-experienced. Bars indicate titer mean and dots reflect the antibody amounts as serum dilution end-point titers. TBEV-vaccinated participants are depicted in orange and TBEV-unvaccinated in blue.

binding for selected EDE. Similarly, the YF17D E locked-dimer construct may have failed to deplete the principal antibody fraction responsible for the virus neutralization (Fig. S4).

sE monomer cannot be used to deplete exclusively monomer-specific IgG antibodies as dimer-specific antibodies may assemble sE monomers together into dimers and therefore, this construct would

also deplete antibodies with dimer-specificities[41]. To ensure the removal of antibodies with monomeric but not dimeric specificities, we then performed subsequent depletions with DI-II and DIII. Even though the depletions resulted in a clear loss of neutralizing capacity, especially in TBEV-experienced individuals, monomeric specificities only made up a minor fraction of the polyclonal neutralizing antibody

**A**

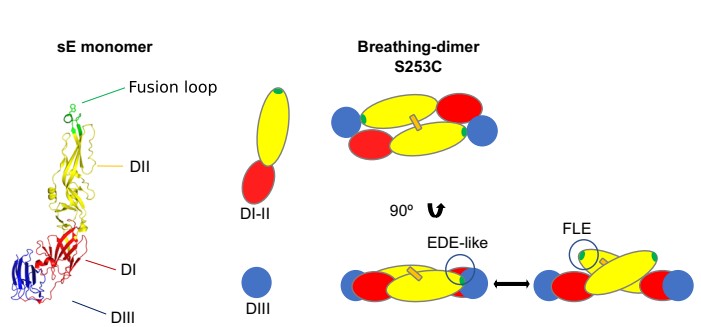

| Antigen | Sequence changes | Expected epitope display |
|---|---|---|
| Monomer sE | - | DI, DII (+FLE), DIII |
| DIII | - | DIII |
| DI-DII | - | DI, DII (+FLE) |
| Breathing-dimer | S253C | EDE-like, DI, DII (+FLE), DIII |
| Breathing-dimer$^{W101H}$ | S253C, W101H | EDE-like(-FL), DI, DII (-FLE), DIII |

**B** **C** **D**

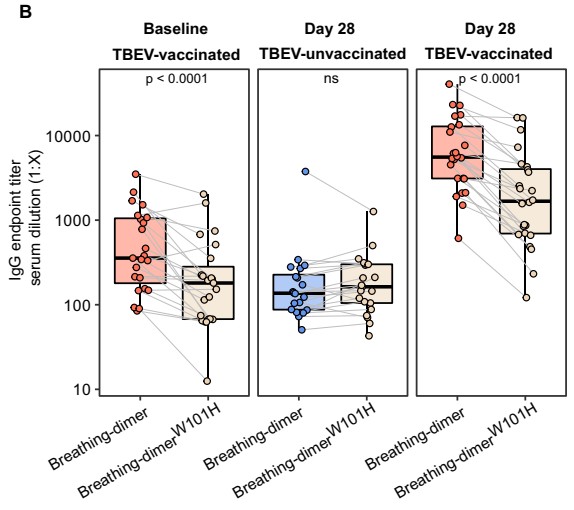

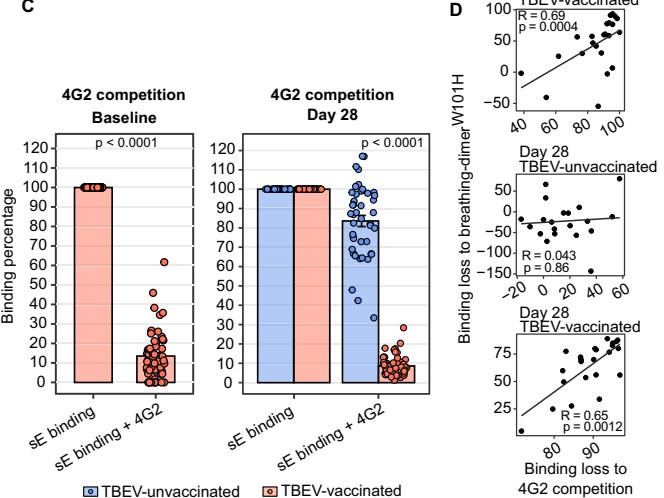

**E** **F** **G**

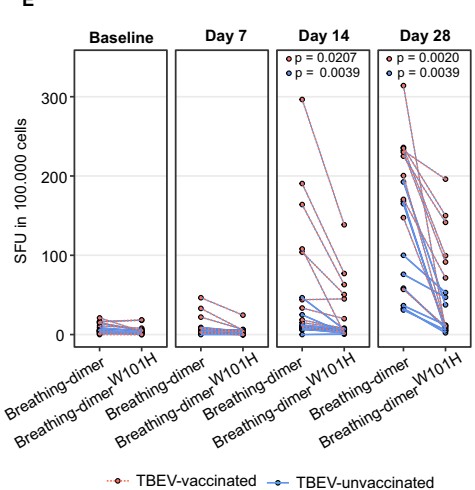

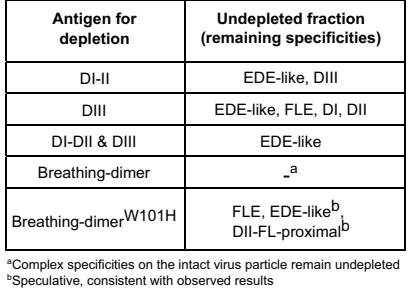

| Antigen for depletion | Undepleted fraction (remaining specificities) |
|---|---|
| DI-II | EDE-like, DIII |
| DIII | EDE-like, FLE, DI, DII |
| DI-DII & DIII | EDE-like |
| Breathing-dimer | _[a] |
| Breathing-dimer$^{W101H}$ | FLE, EDE-like[b] DII-FL-proximal[b] |

[a]Complex specificities on the intact virus particle remain undepleted
[b]Speculative, consistent with observed results

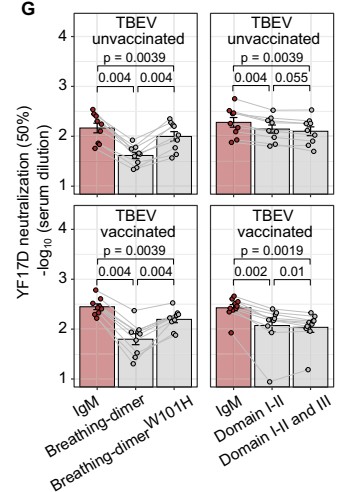

response when compared to the breathing-dimer depleted sera (Fig. 5G right panels).

Altogether, these results highlight the importance of dimer epitopes as the main YFV neutralizing sites and reveal that the FL is a critical component of the binding site for potent neutralizing dimeric antibodies.

## YF17D vaccination boosts antibody-dependent enhancement of DENV and ZIKV infection in vitro in TBEV-pre-immunized individuals driven by FLE antibodies

Given that in TBEV-experienced individuals YF17D boosts a pan-flavivirus cross-reactive IgG response, we examined whether sera from

these individuals mediate ADE of DENV and ZIKV infection using viral reporter replicon particles (VRP)[42].

Interestingly, the antibody response induced by TBEV immunization was sufficient to enhance DENV and ZIKV infection. The enhancing capacity was further increased after YF17D vaccination. As expected, flavivirus-naïve individuals did not facilitate DENV and ZIKV infection in vitro at baseline and the YF17D vaccine did not induce antibodies with enhancing potential. This is consistent with the absence of cross-reactive antibodies in these individuals (Fig. 6A).

To explain the antibody specificities mediating ADE, we first removed the IgM fraction and then measured the enhancing capacity after antigen-specific depletions of the remaining IgG fraction. ADE to

**Fig. 5 | Binding sites and neutralizing capacity of the YF17D vaccine-induced IgG response in TBEV-unvaccinated and pre-vaccinated individuals.**
**A** Recombinant proteins for the dissection and functional analysis of different antibody specificities. The illustration depicts the envelope protein ectodomains (sE), DI-II, and DIII produced separately as well as recombinantly produced dimeric structures. The table summarizes the epitopes displayed by the protein antigens used. **B** IgG endpoint titer quantification for breathing-dimer and breathing-dimer$^{W101H}$ specificities by ELISA at baseline (*n* = 23 TBEV-vaccinated donors) and day 28 (*n* = 24 TBEV-vaccinated, *n* = 20 TBEV-unvaccinated individuals). **C** Antibody binding competition to sE of participants serum with the FL-mab 4G2 at baseline (*n* = 55 TBEV-vaccinated) and day 28 (*n* = 55 TBEV-vaccinated and *n* = 43 TBEV-unvaccinated donors). The percentage of remaining binding is calculated by comparing the binding signal with and without 4G2 as competitor **D** Spearman correlation between antibody binding loss estimated with the 4G2 competition assay (**C**) and with the breathing-dimer$^{W101H}$ (**B**) (*n* = 23 pairs of baseline samples, *n* = 23 pairs of TBEV-vaccinated and *n* = 20 TBEV-unvaccinated of day 28 samples) **E** Longitudinal quantification of IgG-producing B-cells specific for breathing-dimer and breathing-dimer$^{W101H}$ (*n* = 10 TBEV-vaccinated and *n* = 9 TBEV-unvaccinated donors). Units represent spot-forming units per 100.000 PBMC. **F** Table describing the antigens used for antigen-specific IgG depletions and the expected specificities of the remaining undepleted fraction used for YF17D neutralization assays. **G** YF17D neutralization titers (50% cutoff) of IgM-depleted and antigen-specific-depleted sera as explained in **F** (*n* = 9 TBEV-vaccinated and *n* = 9 TBEV-unvaccinated participants). TBEV-vaccinated participants are depicted in orange and TBEV-unvaccinated in blue. Envelope structure accession number (PDB: 6IW4) was edited using Pymol. Individual selection is shown in Supplementary Fig. 3 (SF3). Boxplots display a horizontal line indicating the median and lower and upper hinges corresponding to the first and third quartiles. The lower and upper whiskers extend to 1.5x IQR from the respective hinge. Barplots in **C** and **G** indicate the mean and the error bars the standard error of the mean. Statistical significance between TBEV-vaccinated and unvaccinated individuals (**C**) was estimated with a two-sided Mann-Whitney test. Statistical significance in **B**, **E** and **G** was calculated with a two-sided Wilcoxon signed-rank test. P values above 0.05 are considered non-significant (ns).

DENV was lost in serum depleted of antibodies binding to the breathing-dimer, sE monomer (DI-II and III) or DI-II. In contrast, samples depleted with breathing-dimer$^{W101H}$ or locked-dimer constructs retained antibodies mediating ADE (Fig. 6B, S5F). These results point to FLE-antibodies as responsible for cross-reactivity and ADE.

In conclusion, the YF17D vaccine expands FLE-antibodies with the potential to mediate ADE of DENV and ZIKV infection in vitro in TBEV-pre-immunized individuals, but not in the flavivirus-naïve population.

### Differences in the response pattern between TBEV-pre-immunized and flavivirus-naïve individuals persist for more than one year after YF17D vaccination

The B cell response to the YF17D vaccine continues to mature for 6 to 9 months after vaccination[29]. Likewise, as the immune response advances, the antibody response undergoes diversification in terms of epitope recognition and binding affinity[43]. To extend our findings beyond day 28 post-vaccination, we analyzed serum samples from an independent cohort of 20 individuals collected one year after YF17D vaccination (cohort-2). Although baseline samples were unavailable, a review of the vaccination records identified that 16 individuals had received at least one TBEV vaccine dose before YF17D vaccination, while 4 were TBEV-naïve. Additionally, two individuals who received the YF17D vaccine 9 and 11 years before sample collection, with no record of TBEV vaccination, were part of this cohort (Fig. 7A, B).

Similar to cohort 1, TBEV-pre-vaccinated individuals had significantly higher IgG antibody titers against the YF17D sE protein than TBEV-unvaccinated participants (Fig. 7C). Of note, the difference in titers was approximately one order of magnitude, lower than what we observed on day 28. Moreover, flavivirus-naïve individuals exhibited increased IgG levels compared to those at day 28 (Fig. 7C). Both subgroups efficiently neutralized YF17D one-year pv, with the IgM fraction retaining significant neutralizing potential in participants' sera. Surprisingly, we observed a trend of improved neutralizing titers in TBEV-unexperienced individuals compared to TBEV-pre-vaccinated individuals for both undepleted and IgM-depleted serum (Fig. 7D, E). While the TBEV-unvaccinated sample size is limited, these results suggest that TBEV-pre-vaccinated individuals expanded a high titer of sE-specific antibodies with limited neutralizing potential, while flavivirus-naïve individuals generated a more efficient IgG response for virus neutralization (Fig. 7D, E).

In serum samples from TBEV-pre-vaccinated individuals one-year pv, the sE-IgG binding fraction competed with 4G2 mAb by over 80%. In contrast, flavivirus-naïve individuals retained between 50% and 100% of sE-IgG binding in the presence of 4G2. This result, consistent with the analysis of cohort-1, confirms that the FLE is still a dominant target of the IgG response in TBEV-pre-vaccinated individuals but not in TBEV-unvaccinated donors after one-year pv (Fig. 7F). Moreover, one year after YF17D vaccination, serum from TBEV-experienced donors was still capable of mediating ADE of DENV infection in vitro, while flavivirus-naïve individuals did not elicit an IgG response with enhancing potential (Fig. 7G).

Collectively, these results confirm our observations at day 28 post-vaccination in cohort 1 in an independent cohort sampled one year after YF17D vaccination. In conclusion, TBEV-pre-exposed donors develop a cross-reactive FLE-directed IgG response capable of mediating ADE of DENV infection. However, in flavivirus-naïve individuals, YF17D primes for a non-cross-reactive yet effective neutralizing antibody response.

## Discussion

Flavivirus infections are a major health concern. The global distribution and high prevalence of flaviviruses create the need to understand and integrate the effect of pre-existing immunity to antigenically related flaviviruses in vaccine design and other anti-flavivirus strategies.

In this study, we characterized the antibody response elicited by the YF17D vaccine in a cohort of individuals who were either TBEV pre-immunized or flavivirus-unexperienced. By day 28 post-vaccination, an adaptive response had been mounted, as shown by a 100% seroconversion rate and protective neutralization titers in the study participants of cohort-1. At that timepoint, TBEV-pre-immunity did not influence the polyclonal neutralizing response, which was largely dependent on IgM. The predominant IgM contribution to flavivirus neutralization may be achieved through its multivalent binding to the multimeric epitope arrangement of the virion[44]. Indeed, our results in cohort 2 confirm a prolonged IgM response induced by YF17D that still dominated the neutralizing activity in serum one-year pv and even in the samples 9 and 13 years pv. Notably, the differences observed in the response patterns between TBEV-pre-immunized and flavivirus-naïve individuals at day 28 post-vaccination in cohort 1 were replicated at the one-year mark in cohort 2. By day 28, the IgG fraction contributed to neutralization with higher efficacy in TBEV-pre-immunized individuals, but after one year, despite higher serum IgG levels, TBEV-pre-vaccinated individuals showed a reduced capacity to neutralize YF17D compared to flavivirus-naïve donors. While a larger sample size is needed to confirm this observation, this result may be due to the optimal display of neutralizing epitopes by the YF17D virion and the further maturation of the B cell response between day 28 and one-year[29]. Conversely, TBEV-pre-vaccinated individuals may rely more on pre-existing cross-reactive clones induced after TBEV vaccination, that do not mature further.

Our study shows a robust cross-reactive antibody response induced by the TBEV vaccine, which shares antigenic determinants with the YF17D virus including the FLE (except for a G104H change in

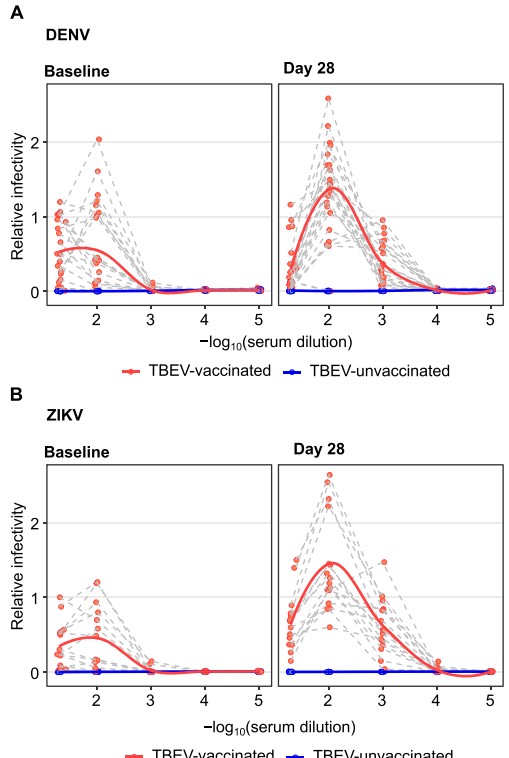

**Fig. 6 | Antibody-dependent enhancement of DENV and ZIKV by bulk and antigen-specific IgG depleted serum in TBEV-unvaccinated and pre-immunized individuals before and after YF17D vaccination. A** Antibody-dependent enhancement of infection with DENV-2 (16681) VRPs at baseline and day 28 post-YF17D vaccination ($n = 23$ TBEV-vaccinated, $n = 15$ TBEV-unvaccinated individuals) **B** Antibody-dependent enhancement of infection with ZIKV (MR-766 African strain) VRPs at baseline and day 28 post-YF17D vaccination ($n = 16$ TBEV-vaccinated, $n = 8$ TBEV-unvaccinated individuals). **C** Dengue ADE for TBEV-vaccinated and unvaccinated individuals driven by: undepleted serum ($n = 22$ per group), IgM-depleted serum ($n = 22$ per group), and, as detailed in Fig. 5F, IgM & antigen-specific-IgG-depleted serum ($n = 9$ per depletion group). Relative infectivity is estimated as the normalized fold-increase of infection to an internal control carried for all the assays. Curves were fitted with local regression. In **A** and **B** TBEV-vaccinated participants are depicted in orange and TBEV-unvaccinated in blue.

the FL sequence)[45]. Pre-existing cross-reactive immunity might thereby influence the subsequent response to the YF17D vaccine in different ways, ultimately contributing to the findings presented here. Firstly, an anamnestic response may occur, explaining the more rapid and robust IgG response towards previously seen epitopes. In this context, pre-existing B cell clones may predominate over de novo cells undergoing germinal center (GC) reactions, thereby limiting the final diversity of the B cell response to YF17D[46]. Nevertheless, we showed that the breadth of the humoral response cannot be explained exclusively by a recall response to conserved epitopes. Secondly, the diversification of antibody specificities can be modulated over time by the presence of antigen-specific antibodies. These antibodies may feedback by either masking or exposing epitopes[47]. In this context, pre-existing cross-reactive antibodies may contribute to broadening the antibody specificities, as we have observed with DIII-specific antibodies. Lastly, we demonstrated that cross-reactive antibodies induced by the TBEV vaccine can enhance YF17D infection via ADE. We showed that higher enhancing titers prior to YF17D vaccine were associated with stronger neutralizing antibody responses within the TBEV-pre-vaccinated individuals suggesting increased vaccine immunogenicity. This is consistent with a previous study by Chan et al (2019) which showed that JEV vaccine-induced cross-reactive antibodies enhance YF17D vaccine immunogenicity via ADE[20]. Further investigation is needed to delineate the interplay between TBEV vaccine imprinting, the feedback of pre-existing antibodies, and the enhanced YF17D immunogenicity mediated by ADE.

Vaccination results in a clear differential IgG response to YF17D, distinguishing flavivirus-experienced from inexperienced individuals at both day 28 and one-year pv. TBEV-experienced individuals had

expanded pan-flavivirus cross-reactive FLE antibodies. In contrast, TBEV-naïve individuals did not develop broader cross-reactivity even after one year but established an effective non-cross-reactive neutralizing antibody response. YF17D virus seems to conceal cross-reactive epitopes while priming for the generation of neutralizing antibodies in flavivirus-naïve individuals. This capacity is not observed in other flavivirus infections or vaccinations that inevitably generate cross-reactive antibodies[10,32,48]. We hypothesize that the specific structure of the YF17D virion leads to limited exposure of the FLE. Thus, the formation of neutralizing antibodies targeting complex quaternary epitopes as well as the FL-proximal region is structurally favored. This particular priming might explain the lack of association between YF17D vaccination and dengue severity[49] contrary to the JEV vaccine which induces a cross-reactive response and is associated with severe dengue disease[18,50]. The expansion of FLE antibodies by TBEV-pre-vaccinated individuals depends on MBC recognition of the epitope. However, the concealment of the FLE by the YF17D virion would hinder this process. Nevertheless, the FLE might still be transiently exposed in the breathing virion enough to trigger a recall response but not enough for a primary response. The FL-dominant response in TBEV-pre-vaccinated individuals may then result from a recall response to a reduced but sufficiently exposed FLE. Alternatively, FLE accessibility could be modulated upon the binding of cross-reactive antibodies to the YF17D virion favoring FLE exposure and dominance.

Extensive literature has described the relevance of the FL-proximal site in DII for YFV neutralization[26–29]. Here, we show that quaternary epitope binders constitute a substantial fraction of the neutralizing activity of the polyclonal human sera post-YF17D vaccination. To date, a precise mapping of EDE-like antibodies for YFV has

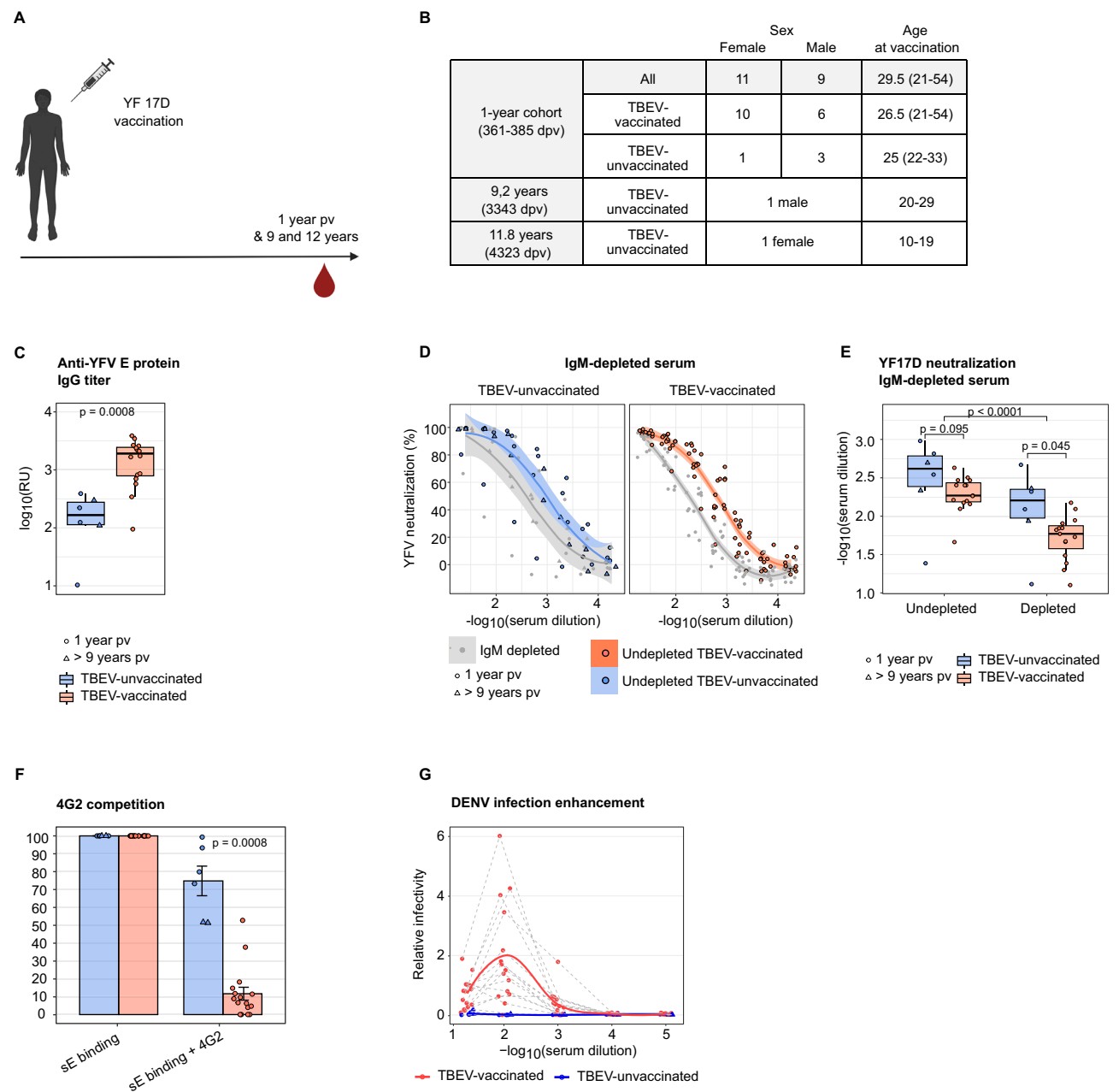

**Fig. 7 | IgG antibody response in cohort-2 sampled one year after YF17D vaccination. A** Diagram representing the serum sample collection of 22 participants. Prepared with Biorender (www.biorender.com). **B** Table summarizing cohort-2 characteristics and TBEV-pre-vaccination status. The table indicates the age at the time of vaccination. **C** YF17D anti-E protein-specific IgG titers in relative units (RU). **D** Neutralization curves of undepleted polyclonal serum and IgG- or IgM-depleted serum (in grey) for TBEV-pre-vaccinated or unvaccinated individuals. **E** Quantification of the 80% neutralization cutoff before and after IgM depletion. **F** Antibody binding competition to sE of participants serum with the FL-mAb 4G2. Percentage of remaining binding is calculated by comparing the binding signal with and without 4G2 as competitor. **G** Antibody-dependent enhancement of infection with DENV-2 VRPs. TBEV-vaccinated participants ($n = 16$) are depicted in orange and unvaccinated ($n = 6$) in blue. One-year pv samples are represented with circles and over 9 years pv with triangles. All individuals are included in every assay except for **D** and **E** ($n = 15$ TBEV-vaccinated donors). Curve fitting in **D** and **G** was calculated with local regression. Boxplots display a horizontal line indicating the median and lower and upper hinges corresponding to the first and third quartiles. The lower and upper whiskers extend to 1.5x IQR from the respective hinge. Barplots in **F** indicate the mean and the error bars the standard error of the mean. Statistical significance between TBEV-vaccinated and unvaccinated individuals (**C**, **E** and **F**) was estimated with a two-sided Mann-Whitney test. Statistical significance between undepleted and IgM-depleted serum samples in E was calculated with a two-sided Wilcoxon signed-rank test. P values above 0.05 are considered non-significant (ns).

not been published. Besides, this characterization could not be done in previous studies when monomeric sE was used for B-cell sorting[28,29]. However, the relevance of EDE-like antibodies in anti-YFV immunity has been previously suggested: monomeric specificities did not completely account for the neutralizing activity in polyclonal human sera[30], and the generation of viral escape variants to potent neutralizing antibodies mapped sites compatible with an EDE-like in YFV[25]. To gain insight into these complex quaternary epitopes, we used covalently bound recombinant dimers with potential to display EDE-like epitopes. We observed a profound loss of neutralizing activity upon removal of

dimer-specific IgG, while sera depleted of IgGs targeting DI-II and DIII largely preserved the capacity to neutralize YF17D. The dissection of antibody specificities in polyclonal serum is challenging given that EDE, FLE and FL-proximal epitopes overlap in some positions in the fusion loop. Nevertheless, our toolbox served to identify dimer epitopes as key determinants of YF17D neutralization.

The efficacy of both TBEV and YF17D vaccines is excellent[24,51]. We show that vaccination against TBEV followed by YF17D did not hamper the immunogenicity and the neutralization response induced by these vaccines. Here we report that sequential TBEV and YF17D vaccinations significantly elevated cross-reactive antibodies that could enhance DENV and ZIKV infection in vitro via ADE. This raises concerns in view of the clinically relevant effect of ADE in dengue disease. Due to ADE, the risk of severe dengue disease is increased in heterotypic secondary DENV infections[15,16]. Additionally, a prior ZIKV infection and a DENV infection followed by a subsequent ZIKV infection[52], and a prior JEV vaccination[18], have been described to elevate the risk for severe dengue disease. Likewise, sequential TBEV and YF17D vaccinations, by increasing non-neutralizing cross-reactive antibodies with enhancing potential, could render individuals susceptible to severe dengue disease. The double vaccination to TBEV and YF17D is not uncommon among European and Asian travelers living in TBEV endemic areas and dengue has become one of the most important emerging diseases among European travelers[53]. Nevertheless, our observations should be interpreted with caution before an epidemiological study evaluates the clinical impact of subsequent TBEV/YF17D vaccination on the severity of dengue disease.

In summary, our study provides valuable insights into the antibody response to the YF17D vaccine. We described how pre-existing immunity influences a response that can be either neutralizing or misdirected to cross-reactive epitopes. This has implications in vaccine design to successfully direct responses to protective antigenic sites with minimal cross-reactivity.

## Methods

### Study samples and personal information

**Cohort-1.** 250 healthy young adults, naïve to natural flavivirus infection and with a negative vaccination history to JEV and YFV, were recruited from 2015–2019 at the Division of Infectious Diseases and Tropical Medicine (DIDTM) as well as the Division of Clinical Pharmacology, University Hospital, LMU Munich, Germany. The study protocol was approved by the Institutional Review Board of the Medical Faculty of LMU Munich (IRB #86-16) and adhered to the most recent version of the declaration of Helsinki. The study cohort was registered in the ISRCTN registry (ISRCTN17974967). 67.6% of the study participants were females ($n = 169$) and the median age at vaccination was 24 years (range 19–47) (Fig. 1C). The body mass index fell in the normal range (18.5 – 24.9) for most participants ($n = 212$; 84.8%) with some outliers ($n = 38$; range 17.7–46.4). None of the participants had fever or symptoms of an acute disease at the time of vaccination. After giving informed consent, longitudinal samples were collected from venous blood draws at baseline (immediately before vaccination) and on days 3, 7, 14 and 28 after subcutaneous YF17D vaccination (standard-of-care administration of 0.5 mL of Stamaril; Sanofi Pasteur, Lyon, France). Samples were analyzed retrospectively and in an anonymized manner. Extended cohort-1 characteristics can be found in Supplementary Table 1 (ST1).

**Cohort-2.** 22 healthy individuals received the YF17D vaccine (standard-of-care administration of 0.5 mL of Stamaril). Participants were recruited at the Mikrobiologisches Institut of the University of Erlangen. The study protocol was approved by the ethics committee of the Friedrich-Alexander-Universität Erlangen-Nürnberg (Az. 350_20B) and adhered to the most recent version of the declaration of Helsinki. With the donor's informed consent, serum samples were collected

approximately one year after vaccination from 20 individuals (361–385 days pv), including 11 females and 9 males, with a median age at the time of vaccination of 29 years. The TBEV pre-vaccination status was determined by examining the vaccination history of the participants, resulting in 16 classified as TBEV-pre-vaccinated and 4 as TBEV-unvaccinated. Additionally, we included two TBEV-unvaccinated participants: one male who received the YF17D vaccine 9.2 years before sample collection and one female vaccinated 11.8 years before collection. Extended cohort-2 characteristics can be found in Fig. 7B.

**Human blood samples.** Blood samples were collected via venipuncture and serum and plasma samples were stored in aliquots at −80 °C. PBMC samples were isolated manually from buffy coats following Ficoll-Paque PLUS (GE Healthcare, Sweden) density centrifugation and cryopreserved in heat-inactivated foetal calf serum (FCS) supplemented with 10% DMSO (Sigma-Aldrich) in liquid nitrogen. All cellular assays started with the thawing of cryopreserved PBMC with an average recovery of 70%.

**Protein cloning, production, and purification.** The YF17D envelope protein ectodomains, subdomains and covalently bound dimers used in this study were produced recombinantly in stably transfected drosophila S2 cell lines (ThermoFisher, R690-07). Site directed mutagenesis (QuikChange II, Agilent, 200523) was performed to introduce a cysteine in position 253 or 107 and 311 to form the covalently bound dimers and a histidine in position 101 to disrupt FLE using the following primers and validated by DNA sequencing.

| W101H | Forward | GATAGAGGCCACGGCAATGGCTGTGGCCTATTTGGG |
|-------|---------|--------------------------------------|
| W101H | Reverse | ACAGCCATTGCCGTGGCCTCTATCAGAATAAGTGCG |
| S253C | Forward | GGAAACCAGGAAGGCTGCTTGAAAACAG |
| S253C | Reverse | GAGCTGTTTTCAAGCAGCCTTCCTGG |
| L107C | Forward | CTGTGGCTGCTTTGGGAAAGG |
| L107C | Reverse | CCTTTCCCAAAGCAGCCACAG |
| T311C | Forward | GAACCCATGCGACACTGGC |
| T311C | Reverse | GCCAGTGTCGCATGGGTTC |

For protein production a pT350 plasmid encoding for the construct of interest with a BiP signal sequence in the N-terminal and a double strep-tag in the C-terminal region was co-transfected with a puromycin resistant plasmid (pCoPuro)[54] with effectene reagent (Qiagen, 301425) following manufacturer's instructions. Cells were then subjected to puromycin selection to generate a polyclonal stable cell line. Protein expression was induced with 5 μM CdCl$_2$ in Insect-XPRESS medium (Lonza) for 5–7 days and supernatants were concentrated and purified using strep-tactin affinity chromatography columns in an AKTA FPLC system. To separate properly folded protein from aggregates, the protein was further purified by size exclusion chromatography (SEC) using a Superdex 200 10/300 column (GE Healthcare) with 10 mM Tris-HCl pH 8 and 150 mM NaCl, and the elution peaks were validated by SDS-PAGE (not shown). Finally, the protein was stored in a NaCl 150 mM, Tris-HCL 10 mM pH 8 solution.

**Biochemical analysis.** Protein purity and dimerization were validated by SDS-PAGE under reducing and non-reducing conditions with DTT (Fig. S6A). Proper folding and epitope display were confirmed upon binding with monoclonal antibodies with defined three-dimensional epitopes like 5A (produced in-house with available public sequences[25,27] in the mammalian expression system ExpiCHO ThermoFisher, specificity against DII and FL-proximal region) and 4G2 (FLE-specific; purified from hybridoma, D1-4G2-4-15 ATCC, HB-112) by ELISA (Fig S6B) and by protein/antibody complex formation visualized in SEC (Fig S6C). ELISA was performed as described below, coating with 1 μg/ml of protein, incubating with serial dilutions of 5A, 4G2, and

strep-tactin-HRP (Bio-Rad, 1610381) as control, and detecting with an anti-mouse-HRP (cell signalling, 7076, 1:1000) or anti-human-HRP (jackson ImmunoResearch, 109-035-088, 1:5000) antibody. SEC runs were performed at 25 °C in a Superdex 200 10/300 GL column at a flow rate of 0.4 mL/min in 10 mM Tris-HCl (pH 8) and 150 mM NaCl. Protein injection concentration was 50-20 µg and the antibody-protein complexes were formed at a 1:2 protein:antibody molar ratio for 30 min at RT before SEC using the monoclonal antibodies 5A, 4G2 or the in-house produced E21.3 (with specificity to DII). Antibody binding to the protein results in a shift in the elution volume of the protein peak due to the increased size of the complex.

**YF17D and YF17D-Venus virus production.** The YF17D virus was directly amplified from a Stamaril vaccine dose. YF17D variant YF17D-Venus plasmid was a generous gift from Charles M. Rice and Margaret MacDonald (The Rockefeller University, New York, USA). Virus stock production and purification were done as previously described[55,56] with small modifications. Briefly, Vero B4 cells (ATCC, CCL81) were expanded (DMEM 10%, 2 mM L-glutamine, and 1% pen/strep) at 37 °C and 5% $CO_2$ and infected at an MOI 0.1. Supernatants were collected post-infection when a clear cytopathic effect was visible. Supernatants were then combined with polyethylene glycol (PEG 8000) 7% (w/v) and the pellet was resuspended and homogenized for final purification by sucrose cushion separation after ultracentrifugation in an MLS 50 rotor (Beckman). The final purified virus stock was diluted in TNE buffer (20 mM Tris-HCl pH 8, 150 mM NaCl, 2 mM EDTA) quantified by plaque assay, and stored at -80 °C until use.

**TBEV plaque reduction neutralization tests (PRNT).** The TBEV reference strain Neudoerfl was produced and titrated as described by Baer & Kehn-Hall 2014[57]. Seven dilutions (1:10–1:640) of heat-inactivated plasma were combined with a constant concentration of TBEV (500 pfu/mL) and incubated for one hour at 37 °C and 5% $CO_2$. Next, confluent A549 cells (ATCC, CCL-185), cultured in MEM, 1x NEAA, 10% FBS medium (Gibco, Thermo Fisher Scientific Inc.) at 37 °C and 5% CO2, were inoculated with the virus-plasma mixture for one hour before the addition of 100 µL CMC medium (MEM, 1x NEAA, 2% FCS, 0.75% carboxymethylcellulose). Following a 3-day incubation, plates were stained with crystal violet (13% formaldehyde, 0.1% crystal violet) at 4 °C, washed, and dried for visual examination. The mean plaque count of every dilution triplicate was determined and compared with the virus-only control to calculate the percentage of plaque reduction. Neutralization curves were fitted by a 4-parameter logistic regression using the drm function of the drc R package. PRNT80 values were interpolated from the curves.

**Indirect immunofluorescence tests (IIFT).** Plasma samples of 39 study participants were tested for IgM as well as IgG reactive to TBEV, WNV, JEV, YFV, and DENV (types 1–4) by using EUROIMMUN Flavivirus Mosaic 3 and ZIKV IIFT assays (EUROIMMUN Medizinische Labordiagnostika AG, Lübeck, Germany) following manufacturer's instructions. For standardized analyses, we used the titerplane technique as described in Niedrig et al. 2008[58] and the automated washing system MERGITE! (Euroimmun AG, Lübeck, Germany). serial two-fold dilutions from 1:10 to 1:5120 were used to determine antibody titers. For antigen-specific IgM detection, samples were prepared with a RF absorbent (EURO-SORB, 2% Tween). A fluorescent microscope (EUROStar 3 PLUS, EUROIMMUN Medizinische Labordiagnostika AG, Lübeck, Germany) was used for the evaluation of specific perinuclear FITC fluorescence for IgM or IgG. Samples with titers below 1:10 were considered negative and samples with a titer of 1:5120 were considered as ≥1:5120.

**Determination of YF17D neutralizing antibody titers.** The neutralizing antibody titer was determined by a Fluorescence Reduction Neutralization Test (FluoRNT) as previously described by Scheck et al, 2022[56]. Briefly, equal amounts of YF17D-Venus virus was incubated with serially-diluted donor sera for 1 h at 37 °C in serum-free DMEM. 21.000 PFU/well of virus was used to reach approximately a 50% cell infection. The mixture was then added to Vero B4 cells (ATCC, CCL81) and incubated for 24 h. Cells were then trypsinized, stained for viability (Thermo Fisher, L10120) and fixed with 4% PFA before acquisition in FACS Canto or CytoFLEX LX. The gating strategy is detailed in supplementary Fig. 1A. Frequency of infected cells in the absence of vaccinee serum was set as 100% and was used as a reference to calculate the percentage of infection reduction. Neutralization curves were fitted by a 4 parameter logistic regression using Prism 8 (GraphPad, La Jolla, CA, USA) or the drm function of the drc R package. 50, 80 and 90 % FluoRNT values were interpolated from the curves.

**Quantification of antigen-specific antibodies in human serum.** An in-house three-layer ELISA was used for the quantification of antigen-specific IgG and IgM in vaccinees serum. Half-area 96 well plates (Corning) were coated overnight at 4°C with 1 µg/mL or 3.5 µg/mL of sE protein or full virus YF-17D respectively (estimated by Bradford assay) in 0.1 M sodium carbonate buffer pH 9.5. For determining anti-DIII IgG responses, 1 µg/mL of recombinant YF17D-DIII or TBEV-DIII (Jena Bioscience, PR-1450-S) was used. Only full virus YF17D was used as the antigen for the quantification of vaccine-specific IgM. Following washes with PBS tween-20 0.05% the plates were blocked with 10% goat serum in PBS+tween-20 0.05% for 3 h at room temperature and constant shaking. Next, heat-inactivated serum samples were added in three-fold dilutions (typically 1:100, 1:300, 1:900 and 1:1800 for IgG detection and 1:200, 1:400 for IgM) in blocking buffer and incubated for 2 h. To ensure the removal of unspecific immunoglobulins, plates were thoroughly washed before the addition of anti-human IgG-HRP (Jackson ImmunoResearch, 109-035-088, 1:5000) or anti-human IgM-HRP (ThermoFisher, 31415, 1:5000). One hour later, plates were washed again before the addition of TMB substrate (BD, 555214). Plates were allowed to develop for 10–25 min before adding 1 M $H_2SO_4$ stop solution. The optical density was read at 450 nm with 595 nm correction in the FLUOstar Omega reader (BMG Labtech). The signal was considered positive if it was at least 3x higher than background measured in wells without pre-coated antigen and the same serum dilution. We excluded data points when the background signal was less than three times lower than the signal in wells with pre-coated antigen, across all tested serum dilutions, or when the background was high that rendered the measurement unreliable. The background signal was subtracted. To ensure inter-day and inter-plate comparability and reproducibility, ELISA titers are reported as relative units (RU) to a standard anti-YF17D serum given an arbitrary value of 10000 units as described previously[32,59,60]. The corresponding values for the test serum samples are read from the standard curve fitted using a four-parameter logistic regression.

**Competition ELISA.** Murine isotypes of the 4G2 (FLE-specific, purified from hybridoma: D1-4G2-4-15 ATCC, HB-112) and 2D12 (DII-specific, purified from hybridoma: 2D12 ATCC, CRL-1689) monoclonal antibodies were employed to compete with human sera for binding to the sE protein. As described above, ELISA plates were coated with 1 µg/mL of sE protein and blocked with 10% goat serum in PBS+tween-20 0.05% and an equimolar concentration (35 nmol) of 4G2 or 2D12 mAb in corresponding wells for 3 h. Samples were tested in 1:100 and 1:1000 dilutions with an additional 35 nmol of 4G2 or 2D12. ELISA titers, reported as RU, were compared to the titer quantified in the absence of any competing antibody.

**Endpoint titer IgG ELISA.** IgG antibody titers directed against locked-dimer, breathing-dimer, and breathing-dimer[W101H] antigens were quantified as endpoint titers. In brief, following coating and blocking of the ELISA plates, seven serial dilutions of the serum samples

(starting 1:50 and diluted 1:3) were added to the plates. Following a 2 h incubation, plates were washed as described above with two additional washes with PBS + NaCl 650 mM. Adding a chaotropic agent removes antibodies with low affinity while preserving the quantification of antigen-specific antibodies and the protein structure. This makes it possible to reliably compare and quantify antibodies targeting antigens that display or disrupt epitopes. The antibody titer is reported as the interpolated dilution of donor serum where the OD value of 0.1 after background subtraction was reached.

**Bulk IgM depletion.** Serum, diluted 1:3 in PBS, was added to anti-human IgM agarose beads (Sigma, A9935) for 1.5 h at RT. Two depletion rounds were conducted, using a bead volume that was twice the volume of the serum. Beads were pre-washed with PBS and supernatants were removed following a centrifugation (5 min 400 $g$). Complete IgM removal was verified by YF17D IgM and IgG ELISA.

**Bulk IgG depletion.** For complete removal of the IgG fraction in serum samples we used Protein G spin plates (Thermo Fisher, 45204) following the manufacturer's instructions with minor modifications. Briefly, 30 μl of each serum sample were mixed with 90 μl of binding buffer (PBS) and added to the corresponding well on the equilibrated protein G plate. Following a 45 min incubation the plate flow-through was reapplied to the purifying plate and incubated again. The flow-through was then saved as "IgG-depleted" serum. Depletions were verified by IgM and IgG ELISA.

**Antigen-specific IgG depletion.** For the depletion of antigen-specific IgG antibodies, we made use of the high-affinity binding of the double strep-tag encoded in the C-terminal end of the recombinant proteins to MagStrep XT beads (IBA, 2-4090-002). Three depletion rounds were performed to achieve complete depletions. In every round, 20 μl of beads were equilibrated in binding buffer (100 mM Tris-HCl + 150 mM NaCl + 1 mM EDTA, pH 8.0) and conjugated with 5 μg of recombinant protein for at least 45 min at 4 °C. Next, unbound protein was removed with a magnetic separator and beads were washed with PBS before adding the serum samples. Usually, IgM-depleted-serum samples were diluted in PBS to a final volume of 160 μl (final dilution 1:7.5) and incubated for 1.5 h at RT in a tube rotator. Depletions were verified by IgG ELISA with the respective E protein used for depletion.

**Antibody-dependent enhancement of YF17D virus.** ADE of YF17D for baseline serum samples was measured using YF17D-Venus infection of THP-1 (ATCC, TIB-202) and K562 cells (ATCC, CCL-243) cultured in RPMI-1640 10%, 2 mM L-glutamine, and 1% pen/strep at 37 °C and 5% $CO_2$. In a 96-well plate, seven 1:3 serial dilutions of every test serum sample, starting with a 1:10 dilution, were combined with YF17D-Venus at an MOI of 1 in uncomplemented RPMI-1640. Following 1 h incubation at 37 °C, the serum-virus mix was transferred to a round bottom 96 well plate with 30.000 cells/well. After two hours of incubation at 37 °C, the cells were washed three times with PBS and then resuspended in R10 medium (RPMI 1640 supplemented with 10% heat-inactivated FCS, 1% penicillin/streptomycin, and 1% L-glutamine) and left to rest for an additional 62 h. Cells were then stained with a viability dye (Thermo Fisher, L10120), fixed with 4% PFA and acquired in FACS Canto. The gating strategy is detailed in supplementary fig. 2A. To account for inter-day and inter-plate variability, every plate carried the same internal control (serum from a TBEV-pre-vaccinated donor) and test samples were normalized against it as follows.

$$\text{Relative infection} = \frac{(\text{Sample}_{\text{infection}} - \text{Infection}_{\text{cells}_{\text{only}}}) * 100}{(\text{Control}_{\text{internal}} - \text{Infection}_{\text{cells}_{\text{only}}})} \quad (1)$$

Next, the enhancing titer was quantified as AUC with all serum dilutions as the independent variable in GraphPad Prism. Quartile grouping selected the top and bottom 25% as individuals with high and low enhancing titers respectively. For controls, IgM and IgG depletions were performed as described above. For FCγR blockade, a combination of 10 μg/ml of the Fc Receptor Binding Inhibitor (Thermo Fisher, 14-9161-73) and anti-CD32 (Biorad, clone AT10, MCA1075) was added to the cells.

**DENV and ZIKV virus ADE.** ADE for DENV and ZIKV infection was performed using previously described DENV-2 (16681) and ZIKV (MR-766 African strain) reporter virus replicon particles (VRP) expressing a Gaussia luciferase[42]. In a 96-well plate, five serial dilutions of every test serum sample (1:20, 1:10², 1:10³, 1:10⁴, 1:10⁵ for cohort samples or 1:50, 1:500, 1:5.000, 1:50.000 for antigen-specific depleted sera), were combined with VRPs at an MOI of 0.5 in uncomplemented RPMI-1640. Following a 1 h incubation at 37 °C, 10.000 K562 cells (ATCC, CCL-243) were added to the serum-virus mix in each well. After two hours of incubation at 37 °C, the cells were washed three times with PBS, resuspended in R10 medium, and then left to rest for an additional 72 h. Supernatants were then collected and stored until measurement. Luciferase activity was quantified in 20 μl of sample using a FLUOstar Omega reader (BMG Labtech). 20 μM of Coelenterazine substrate (Carl Roth, 4094.3) was injected in every well and signal was acquired for 10 s as previously described[61]. To account for inter-day and inter-plate variability, every plate carried the same internal control (serum from a TBEV and YF17D vaccinated donor) and test samples were normalized as follows.

$$\frac{\frac{(\text{Sample} - \text{cells}_{\text{only}})}{(\text{VRP}_{\text{only}} - \text{cells}_{\text{only}})}}{\frac{(\text{Control}_{\text{internal}} - \text{cells}_{\text{only}})}{(\text{VRP}_{\text{only}} - \text{cells}_{\text{only}})}} \quad (2)$$

**B cell EliSpot.** B cell EliSpot was used for the quantification of total IgG-secreting, vaccine-specific and epitope-specific plasmablasts and memory B cells present in PBMC samples. Briefly, ethanol-activated polyvinylidene difluoride membrane EliSpot plates (Millipore, MAIPS4510) were coated overnight at 4 °C with 10 μg/mL with capture anti-human IgG (Sigma Aldrich, I2136) in PBS. After being washed with PBS and blocked with R10 for 1 h, cells were added. Cryopreserved PBMC were thawed and allowed to rest for 1 h in warm R10. For the detection of plasmablasts, 100.000 cells/well were added to the plates without further processing. For the memory B cell EliSpot, PBMC were stimulated with 1 μg/mL R848 (SigmaAldrich, SML0196) and 10 ng/mL recombinant human IL-2 (Biolegend, 791904) for 5 days before harvesting and plating of 50.000–150.000 cells/well. Plates were then incubated at 37 °C 5% $CO_2$ for 20 h. Uncoated wells were used as a negative control and 5.000 cells/well were plated for bulk IgG detection as a positive control. Plates were then washed with PBS + 0.05% Tween-20, followed by an incubation with 1 μg/mL of recombinant sE WT or dimer-mutants or anti-human IgG-HRP in PBS + 0.5% FCS for 2 h at RT. Following thorough washes with PBS + 0.05% Tween-20 and PBS 650 mM NaCl, plates were incubated with StrepTactin-HRP (Bio-Rad, 1610380, 1:5000) which binds the Twin Strep-Tag included in the C-terminus of all the recombinant proteins. Plates were finally washed again with PBS + 0.05% Tween-20 and spots were developed for 10 min with TMB substrate (Mabtech, 3651-10). Plates were rinsed with water and dried before counting the spots with EliSpot Reader ELR04 SR (AIDAutoimmun Diagnostika GmbH, Strassberg, Germany)

## Statistical analyses
Statistical analyses were performed with Prism 8 software (GraphPad) or R and are specified in each corresponding figure caption.

## Reporting summary
Further information on research design is available in the Nature Portfolio Reporting Summary linked to this article.

## Data availability

All data supporting the findings of this study are available within the paper, its supplementary information, and the supplied Source Data file. Further details on the study cohort-1 can be found in the ISRCTN registry (ISRCTN17974967). Source data are provided with this paper. Study protocols are available (in German language) upon request. Source data are provided with this paper.

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

## Acknowledgements

The authors thank Natalie Roeder, Nicole Lichter and Christine Hoerth for technical assistance. The authors also thank Arne Kroidl, Günter Fröschl and Kristina Huber for serving as clinical study investigators. We thank Liz Schultze-Naumburg for support in building up the yellow fever vaccination cohort. We thank Renate Stirner for her assistance with the ELISpot reader. We thank Ritu Mishra and Cell Analysis Core Facility TranslaTUM for their kind help with taking the microscopy pictures for the IIFT assay. We thank Janett Wieseler and Arlen-Celina Lücke for VRP production. We also thank Malena Bestehorn-Willmann and Simon Philipp Stützle for their great support in the implementation of the PRNT assay. We acknowledge the iFlow Core Facility of the University Hospital Munich for assistance with the generation of flow cytometry data. We acknowledge Life Science Editors for editing services. The authors also thank all the cohort participants who voluntarily participated in the study and donated samples. Parts of this work have been performed for the doctoral theses of ASP, FL, SG, EN, MZ, and LL, at the Ludwig-Maximilian University Munich.

## Author contributions

A.S.P. designed and performed experiments, analyzed and interpreted the data, and wrote the manuscript; F.L., S.G., E.N., E.W., M.Z. and L.L. performed experiments and analyzed and interpreted the data; F.D. performed experiments and provided technical assistance; HK provided theoretical assistance and acquired funding; B.M.K. provided critical reagents, acquired funding and helped interpreting the data; G.D. provided critical reagents and helped interpreting the data; M.H. and J.T.S. helped to initiate and set up the cohort-1; S.E. supervised the project and acquired funding; K.S. and E.M.S. acquired funding and set up cohort-2; A.B.K. acquired funding and supervised the project; M.P. helped to initiate the cohort-1, performed vaccinations served as clinical study investigator, acquired funding, and supervised the project; G.B.-S. provided critical reagents, designed experiments, acquired funding and supervised the project; S.R. conceived the study, set up cohort-1, interpreted data, acquired funding, supervised and gave the general direction of the project and wrote the manuscript. All authors critically reviewed the manuscript and approved the final version.

## Funding

## Competing interests

The authors declare no competing interests. This work was supported by FlavImmunity a combined grant of the German Research foundation (DFG) project number 391217598 to SR and ABK and the French National Research Agency (ANR) project number ANR-17-CE15-0031-01 to GBS; by DFG TRR237 grant project number 369799452 to SR and ABK (TRR237 TPB14) and BMK (TRR237 TPA04); by Yellow4Flavi a horizon Europe grant project number 101137459 funded by the European Union to SR, AK GBS, KS; by grants of the iMed consortium of the German Helmholtz Societies to SR; by the Einheit für Klinische Pharmakologie (EKLIP), Helmholtz Zentrum München, Neuherberg, Germany to SR and SE; KS and EMS are supported by the BMBF, project 01KI2013; a Stipend (TI 07.003) by the German Center for Infection Research (DZIF) to FL; grants by the Friedrich Baur Foundation (FBS) to JTS, HK and MP; a Metiphys fellowship of the Medical Faculty of the LMU Munich to MP; by the FöFoLe Program of the Medical Faculty of the LMU Munich to SG, EN, LL, FL, the international doctoral program "iTarget: Immunotargeting of cancer" funded by the Elite Network of

Bavaria to ASP, MZ, SG, LL, EN. The funders had no role in study design, data collection and analysis, decision to publish, or preparation of the manuscript.

## Additional information

**Antonio Santos-Peral** [1], **Fabian Luppa**[2], **Sebastian Goresch** [1], **Elena Nikolova**[1], **Magdalena Zaucha** [1], **Lisa Lehmann** [1], **Frank Dahlstroem**[1], **Hadi Karimzadeh** [1,3], **Julia Thorn-Seshold** [1,4], **Elena Winheim** [5], **Ev-Marie Schuster**[6], **Gerhard Dobler**[7], **Michael Hoelscher** [2,8,9], **Beate M. Kümmerer** [10,11], **Stefan Endres**[1,12], **Kilian Schober** [6,13], **Anne B. Krug** [5], **Michael Pritsch**[2], **Giovanna Barba-Spaeth** [14,15] ✉ **& Simon Rothenfusser** [1,12,15] ✉

[1]Division of Clinical Pharmacology, LMU University Hospital, LMU Munich, Munich, Germany. [2]Division of Infectious Diseases and Tropical Medicine, LMU University Hospital, LMU Munich, Munich, Germany. [3]Department of Veterinary Sciences, LMU Munich, Munich, Germany. [4]Faculty of Chemistry and Pharmacy, LMU Munich, Munich, Germany. [5]Institute for Immunology, Biomedical Center (BMC), Medical Faculty, LMU Munich, Munich, Germany. [6]Mikrobiologisches Institut–Klinische Mikrobiologie, Immunologie und Hygiene, Universitätsklinikum Erlangen, Friedrich-Alexander Universität Erlangen-Nürnberg, Erlangen, Germany. [7]Bundeswehr Institute of Microbiology, Neuherbergstrasse 11, 80937 Munich, Germany. [8]German Centre for Infection Research, Partner Site Munich, 80799 Munich, Germany. [9]Fraunhofer Institute for Translational Medicine and Pharmacology, Immunology, Infection and Pandemic Research, 80799 Munich, Germany. [10]Institute of Virology, Medical Faculty, University of Bonn, 53127 Bonn, Germany. [11]German Centre for Infection Research, Partner Site Bonn-Cologne, 53127 Bonn, Germany. [12]Einheit für Klinische Pharmakologie (EKLiP) Helmholtz Zentrum München German Research Center for Environmental Health (HMGU), Neuherberg, Germany. [13]FAU Profile Center Immunomedicine, FAU Erlangen-Nürnberg, Erlangen, Germany. [14]Institut Pasteur, Université Paris Cité, CNRS UMR 3569, Unité de Virologie Structurale, Paris, France. [15]These authors contributed equally: Giovanna Barba-Spaeth, Simon Rothenfusser. ✉e-mail: giovanna.barba-spaeth@pasteur.fr; simon.rothenfusser@med.uni-muenchen.de

