## [Peer Review File · Nature Communications]

Prior flavivirus immunity skews the yellow fever vaccine response to cross-reactive antibodies with potential to enhance dengue virus infection.Reviewers' Comments:

Reviewer #1:

Remarks to the Author:

This manuscript by Santos-Peral and colleagues report an investigation into how adaptive immunity from prior vaccination with TBEV affect humoral response to live attenuated YF17D vaccination. The authors show that YF17D neutralization activity at D28 after vaccination was mediated mostly by IgM antibodies in both TBEV unvaccinated and vaccinated groups. They showed additionally that the pre-existing cross reactive IgG antibodies could enhance YF17D immunogenicity, possibly through ADE, that enabled earlier production of anti-EDIII antibodies against YF17D; this response appeared specific to YF17D and not TBEV, suggesting earlier expansion of B cell receptor repertoire. They also found that YF17D vaccination in those with prior TBEV vaccination resulted in expansion of flaviviral cross reactive IgG that could potentially enhance other flaviviral infection, such as DENV. The authors concluded that prior flavivirus exposure followed by YF17D vaccination could increase the risk of severe dengue and Zika due to ADE.

This is a detailed study into the humoral response to YF17D vaccination. I very much appreciate the authors attention to experimental detail. There are, however, several areas that require additional attention and perhaps a more nuanced interpretation. These are:

1. A major issue throughout this manuscript is that the authors refer generally to flaviviruses when some of the disease processes, namely ADE, has only be convincingly and epidemiologically seen in secondary dengue. Impact of ADE on other flaviviral infections remain quite hypothetical. Care should be taken not to over-extend on what applies to dengue to other flaviviral infections.
2. that needs to be addressed throughout this manuscript is antibody-dependent enhancement (ADE). Many use ADE to refer to disease enhancement whereas, I believe, the authors refer to enhancement of YF17D and other flaviviral (in the discussion section) infection. The introduction should make this distinction clear to avoid confusion. Indeed, this lack of a more nuanced stance on ADE could have led to the over-interpretation in lines 428-436 as well as the conclusion of the paper. If indeed, TBEV and YF17D vaccinations represent primary and secondary flaviviral infections, respectively, as suggested by the authors in lines 427-428, then their findings would contradict epidemiological findings; tertiary and quaternary DENV infections have mostly been found to be clinically silent (Olkowsky et al, J Infect Dis 2013; 208:1026-1033). This is likely due to expanded repertoire of anti-dengue antibodies and T cells (Ooi and Kalimuddin, Sci Transl Med 2023; 15:eadk1254). The discussion should thus be more nuanced, and the implications of the experimental findings should be limited to TBEV followed by YF17D vaccination; other factors beyond increased uptake through Fc-receptor-mediated entry, such T cells and viral strain differences, which have all not been explored in this study also contribute to severe dengue.
3. Lines 90-95. The description on EDE antibodies are not correctly presented. Antibodies that bind EDE are cross-neutralizing and are mostly produced after secondary DENV infection. They have not been described in those that have only a single flaviviral experience.
4. Line 138. Was TBEV immunity determined by positivity on both PRNT and ELISA or positivity on either PRNT or ELISA?
5. Line 140. I may have missed it, but I was not able to find a description of this ELISA in the methods section. It would also be particularly helpful to know if this ELISA would detect cross-reactive IgG antibodies from prior DENV infection. Indeed, missing in the supplementary tables and text is whether any of the TBEV unvaccinated subjects have had prior DENV infection from travel to dengue endemic regions. Showing either a reference or data that flaviviral cross reactive anti-DENV antibodies would be detected in this ELISA would be particularly helpful.

6. Line 162. Should anamnestic antibody response not also be included as a possibility besides earlier response to YF17D in those with prior TBEV vaccination?

7. The section under the heading in line 196 does not clearly distinguish between the effects of ADE and the effects of anamnestic memory B cell activation. Extended Data Figure 2 shows a linear and not a bell-or inverted V-shaped correlation between baseline IgG titers and YF17D PRNT titer at D28. The one set of data that could argue against anamnestic B cell response is those shown Figure 3D-F. However, most of the other sets of data can be explained by activation of memory B cell from prior TBEV vaccination.

8. Lines 248-265. Is there any data to show that the recombinant proteins accurately represent the conformation on flaviviral envelope? Without such data, it is difficult to interpret findings reported in lines 309-311. Could the loss of binding be due to improper folding of the recombinant proteins?

9. Line 338 and Figure 6. Please state the type of DENV that was used in this experiment.

10. Figures 2b, 4a and 4b. It would be useful to calculate and present the PRNT50 titer. As presented, the data is non-linear and lacks resolution.

11. The extent of cross-reactivity of IgG antibodies following TBEV and YF17D vaccination was only assessed at D28 after YF17D vaccination. Affinity maturation in B cells occur over several months. The quality of the antibody binding and hence extent of ADE could thus change with time from vaccination. This limitation should be discussed.

Minor comments:

1. Line 72. It should be clarified that the estimate of 400 million infections applies to DENV alone, not all of the flaviviruses mentioned in lines 68-70.

2. Line 84. The E protein is the main target of neutralizing antibodies – Other viral proteins, such as prM and NS1 also contain many B cell epitopes.

3. Lines 133-145. Reference should be made to the supplementary tables somewhere in this section.

4. Line 232. Do the authors mean “humoral response to YF17D” rather than “YF17D vaccine response”?

Reviewer #2:

Remarks to the Author:

NCOMMS-23-29635-T: "Prior flavivirus immunity skews the yellow fever vaccine response to expand cross-reactive antibodies, enhancing Zika and dengue virus infection".

The manuscript by Santos-Peral et al. is a study on the antibody response after yellow fever 17D vaccination in naïve and previously TBE vaccinated recipients. Samples from the donors were drawn prior to and on day 3, 7, 14 and 28 after vaccination. The previously TBE vaccinated recipients developed cross-reactive IgG antibodies with a lower capacity to neutralize YF17D reacting to the flavivirus fusion loop epitope (FLE). Serum from the previously TBE vaccinated recipients increased the infection rate for other flaviviruses in vitro. The naïve YFV vaccinated individual developed a non-cross-reacting neutralizing response. By the usage of recombinant YF17D envelope protein mutants the authors further studied the target of neutralizing antibodies after YFV vaccination.

Overall, this article has a nice and systematic way of investigating the antibody responses and binding sites for cross reactive antibodies and neutralizing antibodies after YFV vaccination in naïve and

previously TBE vaccinated study subjects. The data add to the current literature and knowledge. The figures are clear and presented in a logical way. One major issue is the short time after vaccination the samples were drawn (28 days) as the antibody response may mature and wane over time. Thus, a longer timeline and some clinical observations on ADE on other flavivirus infections after TBE (+ YFV) vaccination would be of importance to draw conclusions on the potential increased risk of severe dengue and Zika disease due to ADE. If no samples are available from later time points it should be discussed in more detail. Also for the ADE it needs to be clearly written that it is *in vitro* (also in title) or the authors should provide evidence of clinical cases with ADE after TBE/YFV vaccination. The methodology is sound except for the already mentioned time line that would have benefitted from later time points, as well as the plasmablast experiments.

Please see below for specific comments.

Comments:

1. Figure 1b, what were the ages within each group (pre-TBE vaccinated and non-vaccinated), this should be included in the graph.
2. The TBE vaccination had occurred >4 weeks before YFV vaccination; it would have been interesting to see what exact the time point for the last dose was in the individuals and number of doses.
3. Did the authors compare the viremia in the 2 groups? Did previous TBE vaccination affect the viremia and/or viral clearance in any way?
4. Figure 2F; surprisingly low levels of plasmablasts, however, in this study frozen PBMCs are used in the plasmablast experiments, in general the recovery of plasmablasts in frozen samples are significantly lower and fresh PBMCs should be used for these experiments. In the literature, data show a plasmablast expansion in YFV vaccinated naïve donors peaking around 10-14 days (Wec et al, Kohler et al). Thus these data are in conflict with the literature and might be explained by methodological problems. Flowcytometry staining on the B-cell populations would have added to the study.
5. As the main mediator of neutralization on day 28 was the IgM fraction (75%), later time points at 6-12 months would have added important information to the study.
6. Figure 3 A, also a plot with sera from a YFV vaccinated TBE-neg donor should be shown. It would also be of interest to see how the ADE of YFV vaccinated donors would look at sera from later time points.
7. Figure 4. As mentioned under point 2. The time since the last dose of TBE vaccination would have been of major interest. Also, would this cross reactivity also be valid for other inactivated flavivirus vaccinees (only baseline in this context)? E.g JEV.
8. Would the cross-reactive antibodies decrease as the antibodies affinity mature and the IgG antibodies also wane in serum over time? A later time point preferentially after one year would certainly add to the story. It would also be important to perform the *in vitro* infectious experiments at a later time point to understand if the *in vitro* data on ADE on heterologous flaviviruses still stand.
9. Page 11, line 344 *in vitro* should be included after infection.
10. Discussion page 14, line 428. The authors should add that the enhanced infection is observed *in vitro*.

Reviewer #3:

Remarks to the Author:

In this study, the authors examined the impact of prior vaccination against tick-borne encephalitis virus on the immune response induced by the yellow fever virus vaccine YF17D. Their findings, presented in a very well-written manuscript and supported by a compelling array of assays, show that the IgG response is skewed toward the pan-flavivirus E-Fusion Loop epitope. Flaviviruses are antigenically related, and during natural infection or vaccination, they generate an antibody response that can cross-react with flaviviruses from the same serocomplex as well as from other serocomplexes, as shown in this study. Understanding the complex immune response due to sequential infection is important to assess the risk (symptomatic or severe infection) during

subsequent infection and adapt vaccine strategies.

Comments :

1- If the word count allows, the authors should change the title (Prior flavivirus immunity by Prior TBEV immunity) in order to align it with the scope of the study.

2- Rephrase or omit the last sentence of the abstract. Together, these results suggest that with prior *TBEV vaccination* exposure *to* YF17D *leads to an antibody response skewed toward the FL epitope which can potentially increase the risk of severe

Here the authors do not measure how an increase in anti-FL IgG correlates with the severity risk of subsequent DENV or ZIKV infection. They only show that anti-FL antibodies mediates ADE in vitro. As a result I would suggest mentioning the potential risk without making any assumptions.

3- Supplementary tables do not have a number/title.

4- The authors should include in the results section a more detailed description of the cohort. This should encompass information about the geographical area where the cohort was conducted, a clarification of the inclusion and exclusion criteria applied to the cohort, details regarding the periods of recruitment, and specific dates of data collection.

5- a) The demographics and clinical backgrounds of both groups (TBEV vaccinated vs. non-vaccinated) should be included in Table 1 (the unstratified one), as these represent the two primary groups of interest in the study. It is critical to assess whether there were any significant differences in relevant factors between these two groups. Additionally, if the groups were matched, it is essential to specify the criteria used for matching.

b) Since "fever" is mentioned in the participant clinical data, it would be beneficial to indicate whether participants experienced fever in the 4 weeks preceding YF17D vaccination.

c) I also recommend that the authors streamline the list of clinical information to include only factors that are directly relevant to the study. For instance, it may not be necessary to include information about siblings or children unless there is a clear connection established between these factors and the antibody response after YF17D vaccination.

6- #MAJOR Comment: Please provide a description of the assessment of prior flavivirus immunity (other than TBEV). Include details about the specific flaviviruses tested, the methods employed, and the antigens used.

7- Define the term "baseline." While I assume it refers to the time point immediately before YF17D vaccination, it's important to explicitly specify this definition as it may not be inferred by the reader.

8- #MAJOR Comment: It is crucial for the authors to include p-values in each panel of the figure to provide a clear indication of the significance of the observations

9- Figure 2C "The IgM titer reached a plateau between day 14 and 28 pv and was comparable in both groups of vaccinees" What was the test used for comparing groups before IgG depletion? It looks like Anti-YF17D IgM titers are lower in the TBEV vaccinated vs TBEV un-vaccinated groups without depletion particularly on days 14 and 28. Was this difference significant?

10 - #MAJOR Comment: I strongly recommend that the authors include a supplementary figure displaying the characterization and quality control data for the set of recombinant proteins and domains used in the study. While I believe that the constructs behave as described it is important to provide empirical data to support it. Specifically, it would be valuable to show the binding of a range of monoclonal antibodies, including multiple FL antibodies targeting slightly different epitopes than 4G2, EDE1 and 2, etc

11- line 421 – clarify : "In addition, we show that higher enhancing titers* are associated with stronger neutralizing antibody response..."

*titers prior to YF17D vaccine ?

12- line 435 "the predisposition" – high magnitude of anti-FL IgG titers is not a causal factor. FL-Abs are likely not enough to predispose someone to severe dengue. Please rephrase the sentence

We would like to express our gratitude to the reviewers for their comments and their efforts to enhance the scientific quality of our research article.

Please find below our responses to the specific points raised by the reviewers. Additionally, an updated version of the manuscript has been provided, with the changes highlighted to address the feedback received.

Importantly, we have included an additional figure, Figure 7, which features a set of assays conducted on a separate cohort. The cohort consists of healthy European Caucasians, with serum samples collected one year after YF17D vaccination. The purpose of these experiments was to investigate whether our observations extend beyond day 28 post-vaccination, a point raised by reviewers for further discussion. The manuscript has been updated to include this new dataset. Of note, the discussion has been re-structured to address reviewers' petitions, discuss the observations made with cohort-2 and give a more detailed interpretation of this research in its context. In addition, a supplementary figure with a more detailed characterization of the protein constructs has been added.

Reviewer #1 (Remarks to the Author):

This manuscript by Santos-Peral and colleagues report an investigation into how adaptive immunity from prior vaccination with TBEV affect humoral response to live attenuated YF17D vaccination. The authors show that YF17D neutralization activity at D28 after vaccination was mediated mostly by IgM antibodies in both TBEV unvaccinated and vaccinated groups. They showed additionally that the pre-existing cross reactive IgG antibodies could enhance YF17D immunogenicity, possibly through ADE, that enabled earlier production of anti-EDIII antibodies against YF17D; this response appeared specific to YF17D and not TBEV, suggesting earlier expansion of B cell receptor repertoire. They also found that YF17D vaccination in those with prior TBEV vaccination resulted in expansion of flaviviral cross reactive IgG that could potentially enhance other flaviviral infection, such as DENV. The authors concluded that prior flavivirus exposure followed by YF17D vaccination could increase the risk of severe dengue and Zika due to ADE.

This is a detailed study into the humoral response to YF17D vaccination. I very much appreciate the author's attention to experimental detail. There are, however, several areas that require additional attention and perhaps a more nuanced interpretation. These are:

1. A major issue throughout this manuscript is that the authors refer generally to flaviviruses when some of the disease processes, namely ADE, has only been convincingly and epidemiologically seen in secondary dengue. The impact of ADE on other flaviviral infections remains quite hypothetical. Care should be taken not to over-extend on what applies to dengue to other flaviviral infections.

Response: We agree with the reviewer's point and acknowledge that ADE of disease has only been convincingly proven for dengue (e.g., Katzelnick et al. Science 2017). Acknowledging that the evidence for enhanced Zika disease is insufficient and that ADE effect on flaviviruses other than dengue is hypothetical, we corrected the text accordingly, removing from the text every instance when Zika disease was associated to ADE.

- Abstract lines 47 and 54 we made explicit ADE of infection was measured in vitro
- Discussion lines 481-496

2. that needs to be addressed throughout this manuscript is antibody-dependent enhancement (ADE). Many use ADE to refer to disease enhancement whereas, I believe, the authors refer to enhancement of YF17D and other flaviviral (in the discussion section) infection. The introduction should make this distinction clear to avoid confusion. Indeed, this lack of a more nuanced stance on ADE could have led to the over-interpretation in lines 428-436 as well as the conclusion of the paper.

Response: We understand this point and we therefore stressed that we refer to antibody-dependent enhancement of infection since we do not measure disease or clinical outcomes of disease. To this aim, we have made the following changes:

- Abstract: in line 54 we add *in vitro* to make clear the ADE of dengue and Zika infection refers to an *in vitro* assay and not a clinical outcome. We also changed the last concluding sentence of the abstract to a more cautious interpretation of this study's implications.
- The introduction was changed in lines 106 to 107 in an attempt to make the distinction between ADE of infection and ADE of disease clearer
- Lines 130-132 (end of the introduction). dengue and Zika were corrected to DENV and ZIKV to more accurately indicate we refer to virus and not disease
- In line 205 212, 228, 230, 237 we made more explicit that TBEV-induced IgGs mediate enhancement of YF17D infection. Using the expression "ADE of YF17D infection"
- In 342-362 we clarify that we measure ADE of DENV and ZIKV infection *in vitro*. We consequently replace dengue and Zika with DENV and ZIKV to clarify it is the virus and not the disease that we are referring to as we comment on the enhancement of infection.
- The final part of the discussion has been changed to a more cautious interpretation of our findings in the context of current knowledge of ADE and dengue disease.

If indeed TBEV and YF17D vaccinations represent primary and secondary flaviviral infections, respectively, as suggested by the authors in lines 427-428, then their findings would contradict epidemiological findings; tertiary and quaternary DENV infections have mostly been found to be clinically silent (Olkowsky et al, J Infect Dis 2013; 208:1026-1033). This is likely due to expanded repertoire of anti-dengue antibodies and T cells (Ooi and Kalimuddin, Sci Transl Med 2023; 15:eadk1254). The discussion should thus be more nuanced, and the implications of the experimental findings should be limited to TBEV followed by YF17D vaccination; other factors beyond increased uptake through Fc-receptor-mediated entry, such T cells and viral strain differences, which have all not been explored in this study also contribute to severe dengue.

Response: We understand the reviewer's point. Although tertiary and quaternary DENV infections are indeed clinically silent, we think that subsequent TBEV and YF17D vaccinations do not represent the scenario resulting from a primary and secondary DENV infection. We base this interpretation on the following points:

- The findings by Katzelnick et al, Science 2020, showed that "DENV infection followed by one ZIKV infection also increased the future risk of dengue disease, unlike for sequential DENV infections, which reduced future risk". Antibody levels elicited after a single ZIKV infection are also associated with worsened clinical dengue disease. This finding suggests that subsequent infections with heterologous viruses (like ZIKV/DENV) do not have the same effect as sequential infections with different DENV serotypes which are likely to broaden the BCR repertoire diversity towards cross-neutralizing determinants rather than boosting cross-reactivity with ADE potential.
- Secondly, to our understanding, there are no known shared cross-neutralizing determinants between yellow fever and dengue or TBEV and dengue, which could favor clinically silent dengue infections. Instead, there is a minimal (or null) cross-reactivity between YF17D first and subsequent dengue (this study, Nathalia Caroline Santiago e Souza, IJID, 2019) Also, Luppe et al. (Travel Medicine and Infectious Disease 2019) showed no association between YF vaccination and the course of dengue disease. Therefore, we argue that the subsequent TBEV and YF17D vaccination, and not YF17D vaccination alone, would result in a humoral response comparable to that after a primary DENV infection.

We agree with the reviewer that previous studies have contemplated the impact of cross-reactive T cells in ADE of disease although with conflicting results and without a clear conclusion on what would happen clinically in humans. Either cross-reactive CD4 T cells mitigate ADE of disease, potentially by improving antibody avidity and maturation and then contributing to protection to a heterotypic challenge or even by contributing to viral clearance (Chen JEM 2020) or they may contribute to pathology, potentially by increasing immunopathology or promoting an original antigenic sin in the humoral response. It is true that we do not evaluate in this manuscript the T cell response elicited upon YF17D vaccination for potential cross-reactivity against DENV. It should be noted that TBEV vaccine is an inactivated vaccine which therefore may only induce a T cell response against structural proteins

To conclude, the point raised by the reviewer that the discussion should be more nuanced, and the implications of the experimental findings limited to TBEV followed by YF17D vaccination is valid. We have restructured the discussion with this in mind, we hoped to avoid overinterpretation and to not extend our conclusions to other flaviviral exposures. Nevertheless, we did not comment on T cells in the interest of conciseness and to avoid digressing into a complicated branch of adaptive immunity that is out of the scope of this manuscript.

3. Lines 90-95. The description on EDE antibodies are not correctly presented. Antibodies that bind EDE are cross-neutralizing and are mostly produced after secondary DENV infection. They have not been described in those that have only a single flaviviral experience.

Response:

It is true that EDE, or EDE-antibodies, have not been identified for YFV as we have commented in the introduction (l.100, l118-119) and in the discussion (l 465-480). Crystal structures of antibodies binding quaternary dimeric structures have been resolved for DENV and ZIKV. Given the absence of glycosylation on the vaccination strains of YFV we also cannot use the established nomenclature of EDE1/2 for YF either. As the reviewer highlights, and since an EDE has not been mapped for yellow fever and it is not the proper nomenclature, we have rephrased throughout the manuscript every instance in which we used the term EDE. Several editions were done in the text and figures to refer to them more appropriately. We have changed the introduction (lines 90-100) where EDE was presented to improve clarity. Throughout the manuscript and figures we have used terms like: “quaternary dimeric epitopes”, “equivalent to EDE in YFV”, “dimeric epitopes”, “antibodies with dimeric specificities” “EDE-like” “sites compatible with an EDE in YFV”

Generally, with our assays, we have identified the presence of antibodies binding the dimer conformation. We believe that our observation that a relevant fraction of the neutralizing response to YF17D is targeting dimeric quaternary epitopes adds novelty to our study. We speculate that YF17D is specifically and preferentially priming B cells for this type of epitope and that an EDE-like exists for YFV. Before us, other studies have identified antibodies which binding site is compatible with EDE: E.g. Daffis et al jv 2005, generated escape mutants to a highly neutralizing monoclonal antibody (7A) and identified the escape mutations located in opposing E protomer units. Likewise, the same aminoacidic positions were mutated in escape mutants to other neutralizing antibodies (Lobigs et al 1987 and Ryman et al 1997). Also, monomeric specificities did not completely account for the neutralizing activity in polyclonal human sera Vratskikh et al PlosPath. 2013.

The reviewer mentioned that EDE antibodies cross-neutralize DENV serotypes and are elicited after secondary exposures. It is true that further B cell diversification after a secondary challenge primes for the generation of EDE antibodies, which in DENV immunology are potent cross-neutralizers. However, in some instances, EDE antibodies have also been found in individuals after a primary DENV infection: Dejnirattisai et al. Nat. Immunol 2015 did find a substantial dominance of EDE antibodies in an individual with a primary DENV infection (patient 752 in this study). Therefore we believe that EDE antibodies can also be generated after a primary flaviviral infection or vaccination. We believe that our identification of an important fraction of the neutralizing response in sera requiring dimeric arrangements for binding could motivate the search and identification of

monoclonal antibodies targeting the dimer for antibody therapies or improving the understanding of neutralization mechanisms.

4. Line 138. Was TBEV immunity determined by positivity on both PRNT and ELISA or positivity on either PRNT or ELISA?

Response: TBEV immunity was decided primarily by a TBEV PRNT and the self-reported pre-vaccination status. This decided the grouping of 233 individuals (93% of the study cohort). The ELISA was used as a confirmatory test. 4 TBEV-vaccinated TBEV-neutralizers did not show antibodies in the ELISA and were excluded

The reviewer can now find in the Source Data Excel Sheet that we provide with the raw data, a more detailed indication of the criteria used to classify each individual (Fig1_TBEVgrouping)

We acknowledge this criterion is strict, and we could have easily inferred the correct TBEV-pre-vaccination status of some of the individuals classified as “unknowns”. Nevertheless, we decided on a very strict grouping to minimize the chances of allocating individuals to the incorrect group.

As part of this investigation, we also have baseline IgG data for YF17D sE and YF17D virus. Even though these data were not considered directly for grouping individuals, it validated the correct allocation into groups showing only cross-reactive antibodies in the baseline for TBEV-pre-vaccinated donors.

5. Line 140. I may have missed it, but I was not able to find a description of this ELISA in the methods section. It would also be particularly helpful to know if this ELISA would detect cross-reactive IgG antibodies from prior DENV infection. Indeed, missing in the supplementary tables and text is whether any of the TBEV unvaccinated subjects have had prior DENV infection from travel to dengue endemic regions. Showing either a reference or data that flaviviral cross-reactive anti-DENV antibodies would be detected in this ELISA would be particularly helpful.

Thank you for pointing out a mistake in the heading of the ELISA section “Quantification of anti-YF17D antibodies in human serum.” Which now is renamed as “Quantification of antigen-specific antibodies in human serum”. The way this ELISA was performed is identical to the sE-YF17D-specific IgG ELISA but using TBEV-DIII antigen instead that we acquired recombinantly from Jena Bioscience (PR-1450-S). The description of the assay can be found in methods.

About potential infection with DENV or other flaviviruses. From the 250 individuals recruited in our cohort, only 24 had traveled to an endemic area (e.g. Brazil, Thailand, etc.). Although we cannot be sure that other natural infections have not occurred, we consider them unlikely. Besides, with the experimental validation on the groups we have done, we can be sure that those grouped as TBEV-unvaccinated did not have any other flaviviral infection that had resulted in cross-reactivity to YFV (as we see in Figure 2E). The TBEV-DIII-specific ELISA, since DIII lacks FLE, is very specific for TBEV-immunity. The provider did not test for cross-reactivity, although mosquito- or tick-borne flavivirus serocomplexes can be well discriminated with DIII ELISA (Holbrook et al J Clin Microbiol. 2004) In the TBEV-negative group, we confirmed the absence of anti-TBEV antibodies and neutralizing capacity. We did not perform a specific test for DENV pre-infection (except for a fraction of individuals that were tested in the IIFT) but we are sure of the lack of cross-reactivity to YF in baseline samples. Concerning the TBEV-pre-vaccinated group, we are certain of their TBEV vaccination history. However, we cannot definitively rule out the possibility of additional natural infections with other flaviviruses (including TBEV). Nonetheless, we consider this possibility to be unlikely and, if it did occur, it would likely affect only a very small fraction of our cohort that have traveled to endemic areas. For this reason, we refer to this cohort as 'naïve to natural flavivirus infection' in our methods section.

In the supplementary table 1 with the characteristics of cohort 1, we have included the travel history to endemic areas for all and TBEV-pre-vaccinated and TBEV-unvaccinated groups.

6. Line 162. Should anamnestic antibody response not also be included as a possibility besides earlier response to YF17D in those with prior TBEV vaccination?

Yes, it is true and this effect is likely an anamnestic response. In this line, we say “TBEV-pre-immunized individuals have cross-reactive IgG antibodies to YF17D and experience an earlier and stronger IgG response” as a description of the observed result. In the now rewritten discussion, we have given more emphasis to the anamnestic response that characterizes the TBEV-vaccinated population: line 427. “an anamnestic response may occur, explaining the more rapid and robust IgG response towards previously seen epitopes. In this context, pre-existing B cell clones may predominate over de novo cells undergoing germinal center (GC) reactions, thereby limiting the final diversity of the B cell response to YF17D”

7. The section under the heading in line 196 does not clearly distinguish between the effects of ADE and the effects of anamnestic memory B cell activation. Extended Data Figure 2 shows a linear and not a bell-or inverted V-shaped correlation between baseline IgG titers and YF17D PRNT titer at D28. The one set of data that could argue against anamnestic B cell response is those shown Figure 3D-F. However, most of the other sets of data can be explained by activation of memory B cell from prior TBEV vaccination.

The reviewer is right, and we can not rule out the possibility that these observations result from the expansion of memory B cells elicited upon TBEV vaccination.

Nevertheless, with this section, we wanted to highlight an alternative possibility, that may be happening in concert with memory reactivation, and that has been shown in literature to take place for live vaccines and, specifically, for YF17D (Chan et al. Nature Microbiology 2019 and reviewed by Mok et al viruses 2020).

We have restructured the discussion and now we discuss in more detail the mechanisms behind the phenotype we observed for TBEV-vaccinated donors (lines 423-444)

We believe it may result from a combination of three factors acting in concert: 1) an anamnestic response by memory B cells to previously seen epitopes. 2) ADE of YF17D infection resulting in increased immunogenicity, 3) antibody feedback, in which pre-existing antibodies may mask and expose alternative epitopes (e.g. making FLE more accessible and favoring Ab diversification to target DIII) and memory B cells that can lower the activation threshold for new B cells in GC (Schaefer-Babajew et al Nature 2022; Hägglöf et al cell 2023)

It is true, that a bell shape in Extended Figure 2 would have given a clearer indication of an antibody-mediated effect via ADE of the outcome of vaccination. Nevertheless, the bell shape defines the window of antibody concentration in which the enhancement takes place and, in our assays, the peak of enhancement occurs at the lowest dilution tested (1:20) Figure 3 A and B. Therefore the linear correlation in conjunction with our controls (FcR-Blockage and IgG-depleted) make the effect likely mediated via ADE. The results section has been re-checked to avoid assertive claims and give a more cautious interpretation of the results

8. Lines 248-265. Is there any data to show that the recombinant proteins accurately represent the conformation on flaviviral envelope? Without such data, it is difficult to interpret findings reported in lines 309-311. Could the loss of binding be due to improper folding of the recombinant proteins?

We have now added an additional supplementary figure 6 with more data that includes a characterization of the recombinant proteins used in this study.

All the proteins were purified by affinity chromatography (using the double strep-tag) followed by size exclusion chromatography (SEC). Typically, protein eluted in different peaks (corresponding to misfolded/ aggregated or properly folded protein). The peak containing properly folded protein was validated by gel.

We would like to highlight that some of the constructs engineered for this study have been described before for the wild type YF Asibi. The monomer sE protein was produced and purified as we have published before (Crampon et al. mBio 2023) and the proper folding is optimal, as proven by X-ray crystallography and biochemical assays. The breathing-dimer, which locks with a single mutation the protein as a head-to-tail dimer as on mature virions, was done previously for DENV (at position 259 in sE from DENV1, 2, and 4, and position 257 in DENV3) (Rouvinski, NatComm 2017). The position S253 equals that on DENV and allows the formation of a head-to-tail dimer in YFV. Importantly, we had already published the beathing-dimer for YFV Asibi (Crampon et al. mBio 2023. FIG 3 and S3). In this study, the reviewer can find biochemical and structural data on the YF envelope protein. The locked-dimer construct reproduces the strategy implemented by Slon-campos Nat Imm 2019 for ZIKV and Rouvinski et al Nat Communication 2017 for DENV. The positions T311 and L107 in YFV equal those changed for the locked dimer in ZIKV (**L107C**, A264C and **A319C**) and DENV2 (L107C/A313C) DENV3 (L107C/S311C) and DENV4 (L107C/A313C). Lastly, the point mutation W101H was used before for DENV (Klein Jvirol 2013) and TBEV (Meditis EMBO Rep. 2020) to improve the solubility of the recombinant envelope protein. They showed crystallographically how the structure was not changed by this point mutation although W101H prevents insertion into membranes and can be protonated at low pH. Klein 2013 shows that the FL is in a typical orientation and overlaps with other published structures with WT FL sequence. Therefore, the point mutation W101H is not expected to alter the structure or conformation of the protein but it serves to disrupt the epitope.

To further validate the quality and proper folding of the constructs used in this study we have performed additional experiments and we provide further details in the method section. In the supplementary figure 6 the reviewer can find a Coomassie gel illustrating the purity of the recombinant protein under reducing and non-reducing conditions (with and without DTT). In the non-reducing condition, dimeric constructs run as a higher band, corresponding to their molecular weight, and in the reducing condition, all the constructs run as the ~50KDa of the sE monomer. We additionally proved the proper folding of the constructs by testing binding with monoclonal antibodies. In ELISA, we show binding of the sE monomer and the breathing-dimer to mAb 4G2 (FL-specific) and 5A (which binds the FL-proximal area in DII and which epitope has been structurally determined by Lu et al. Cell Reports 2019). As expected, breathing-dimerW101H and locked-dimer have lost the capacity to bind to 4G2 and 5A in the ELISA (Supplementary Figure 6 B). More elegantly, we evaluated antibody-protein binding in their natural soluble conformation by SEC (Supplementary figure 6C). SEC allows the visualization of protein-antibody complexes formation as they elute through a Superdex 200 10/300 column (detailed in methods). In SEC, larger macromolecules or protein complexes elute first. Concerning the Envelope protein, we have experienced that the extended structure of E allows the interaction of the fusion loop with the column which makes the protein elute later than what would correspond to a globular protein of the same molecular weight (Crampon et al mBio 2023). This explains the different elution peaks of sE monomer and breathing-dimer when compared to W101H mutated construct that no longer interact with the column. In the Supplementary figure 6C, all constructs bind the in-house E21.3 mAb that binds to domain II (the exact binding sites are not yet identified). For the breathing-dimer, breathing-dimerW101H and locked-dimer there is a clear shift in the antibody-protein complex to be eluted in an earlier fraction when compared to antibody and protein alone. For sE-monomer/E21.3, complex binding is proven by the lack of protein eluted between 15-20ml range which would correspond to unbound sE protein (red chromatogram). The interaction of the FL with the column explains that the sE-E21.3

complex elutes at the same fraction as E21.3 antibody only. In the same figure, 4G2 and 5A show binding to sE monomer and breathing-dimer but not to breathing-dimer^{W101H} and locked-dimer. Both 4G2 and 5A antibodies bind to a tridimensional epitope, which is only present in properly folded protein. The FLE structure requires a disulfide bond C74 and C105. Consequently, 4G2 does not bind to denatured protein. The FL is directly mutated in both the breathing-dimer^{W101H} and locked-dimer constructs, which explains the lack of binding to 4G2. W101 is part of the 5A epitope (W101 interacts with Y54 of the heavy chain). This explains why the breathing-dimer^{W101H} does not interact with this antibody. Concerning the locked dimer, there is no direct interaction of L107 with the 5A antibody but we could speculate that the locked dimer does not allow access to W101 and this may affect the 5A binding.

Altogether, the protein preparation was performed according to validated procedures that our lab uses routinely for structural studies and our biochemical characterization shows that the constructs have the expected characteristics of size and binding. Furthermore, our depletion experiments are only justified by a properly folded protein. The major depletion of the neutralizing capacity of serum using the breathing-dimer constructs could only be achieved by the proper display of epitopes. Therefore, even though we have not solved the structures of these constructs, we are confident of their suitability for this study.

9. Line 338 and Figure 6. Please state the type of that was used in this experiment.

Response. It is DENV2 (16681) and ZIKV African MR-766 strain. We have added this information to the method section and Figure 6 and 7 caption

10. Figures 2b, 4a and 4b. It would be useful to calculate and present the PRNT50 titer. As presented, the data is non-linear and lacks resolution.

Response: PRNT data of Fig 2b was recalculated to 80% Nt values by fitting a curve as now defined in methods. In a few cases, for the longest dilution tested (1:640) the neutralizing value was higher than 80%, which did not allow the calculation of the Nt value. For those, a dilution value of 640 was given as a PRNT80 value. We did 80% instead of 50 for consistency with the cutoff we used for YF.

Fig4a and b, The IIFT is a semiquantitative assay based on a microscopy readout. Therefore the values cannot be quantified in a continuous unit.

11. The extent of cross-reactivity of IgG antibodies following TBEV and YF17D vaccination was only assessed at D28 after YF17D vaccination. Affinity maturation in B cells occur over several months. The quality of the antibody binding and hence extent of ADE could thus change with time from vaccination. This limitation should be discussed

Response: We completely agree, to complement our findings we got the opportunity to analyse a separate cohort sampled one year post YF17D vaccination. A chosen core of assays was implemented in these new samples. The results are in a new figure we wish to include in the manuscript (Figure 7) and the results have also been commented in the discussion. Although this new cohort has a low sample size, and unfortunately baseline samples were not available, we could confirm the findings we have presented in the first submission of this article. Please find these results in figure 7 and in the text under the heading "Differences in the response pattern between TBEV-pre-immunized and flavivirus-naïve individuals persist for more than one year after YF17D vaccination".

Minor comments:

1. Line 72. It should be clarified that the estimate of 400 million infections applies to DENV alone,

not all of the flaviviruses mentioned in lines 68-70.

Response: It has been clarified in line 72.

2. Line 84. The E protein is the main target of neutralizing antibodies – Other viral proteins, such as prM and NS1 also contain many B cell epitopes.

Response: It has been corrected in the text

3. Lines 133-145. Reference should be made to the supplementary tables somewhere in this section.

Response: It has been added. One table was added to Figure 1 and the extended table as supplementary table 1. In methods, we have also referred to the public repository where more information on this cohort can be found.

4. Line 232. Do the authors mean “humoral response to YF17D” rather than “YF17D vaccine response”?

Response: It has been corrected in the text

Reviewer #2 (Remarks to the Author):

NCOMMS-23-29635-T: "Prior flavivirus immunity skews the yellow fever vaccine response to expand cross-reactive antibodies, enhancing Zika and dengue virus infection".

The manuscript by Santos-Peral et al. is a study on the antibody response after yellow fever 17D vaccination in naïve and previously TBE vaccinated recipients. Samples from the donors were drawn prior to and on day 3, 7, 14 and 28 after vaccination. The previously TBE vaccinated recipients developed cross-reactive IgG antibodies with a lower capacity to neutralize YF17D reacting to the flavivirus fusion loop epitope (FLE). Serum from the previously TBE vaccinated recipients increased the infection rate for other flaviviruses in vitro. The naïve YFV vaccinated individual developed a non-cross-reacting neutralizing response. By the usage of recombinant YF17D envelope protein mutants the authors further studied the target of neutralizing antibodies after YFV vaccination.

Overall, this article has a nice and systematic way of investigating the antibody responses and binding sites for cross reactive antibodies and neutralizing antibodies after YFV vaccination in naïve and previously TBE vaccinated study subjects. The data add to the current literature and knowledge. The figures are clear and presented in a logical way. One major issue is the short time after vaccination the samples were drawn (28 days) as the antibody response may mature and wane over time. Thus, a longer timeline and some clinical observations on ADE on other flavivirus infections after TBE (+ YFV) vaccination would be of importance to draw conclusions on the potential increased risk of severe dengue and Zika disease due to ADE. If no samples are available from later time points it should be discussed in more detail. Also for the ADE it needs to be clearly written that it is in vitro (also in title maybe) or the authors should provide evidence of clinical cases with ADE after TBE/YFV vaccination. The methodology is sound except for the already mentioned time line that would have benefitted from later time points, as well as the plasmablasts experiments.

Response: We thank the reviewer for the nice comments and the detailed reading of the manuscript. In this paragraph, the reviewer has brought up two topics for consideration.

→The short time after vaccination the samples were drawn.

We completely agree, to complement our findings we could use a separate cohort sampled one year post YF17D vaccination. A chosen core of assays was implemented in these new samples. The results

are in a new figure we wish to include in the manuscript (Figure 7) and the results have also been commented in the discussion. Although this new cohort has a low sample size, and unfortunately, baseline samples were not available, we could confirm the findings we have presented in the first submission of this article. Please find these results in Figure 7 and in the text under the heading “Differences in the response pattern between TBEV-pre-immunized and flavivirus-naïve individuals persist for more than one year after YF17D vaccination.”

→ ADE of DENV and ZIKV infection was performed in vitro.

It is correct, we have only measured ADE of DENV and ZIKV in vitro. Throughout the manuscript, we have made more explicit that this assay was in vitro, including the abstract. Following also other reviewer’s comments, we have made clear that we talk about ADE of infection and not of disease. Besides, we have clarified that the clinical association of ADE with disease severity has only been epidemiologically proven for dengue disease. The discussion has been changed to give a more detailed interpretation of our ADE results in the current context and to avoid overinterpretation: 480-495.

We tried to gather clinical data that could show that subsequent TBEV and YFV vaccination may be associated clinically with dengue disease. However, to our knowledge, this data does not exist. The Robert Koch Institute in Germany and the European Center for Disease Control report annually the number of dengue cases in Europe per country. Unfortunately, the severity of the disease is not noted and neither is patient data that we could use to infer previous vaccinations received. Besides, there is an inherent bias in these reports since only clinically apparent dengue may be diagnosed and not all European citizens from all countries travel in the same number to dengue-endemic areas and not all the countries may have the same capacity to screen and diagnose dengue. One stand that we make with this study is the need to conduct an epidemiological study that evaluates if TBEV/YFV vaccination has a clinical implication on dengue severity (see line 494-496 in the discussion). Considering the findings of others like Anderson PNTD 2011, which showed an association between JEV vaccination and Dengue disease and Katzelnick et al, Science 2020, showing antibody levels elicited after a single ZIKV, DENV or ZIKV/DENV infections are also associated with worsened clinical dengue disease together with the data that we present here, we believe there is substantial ground to make this hypothesis.

→ Since we cannot provide clinical data, we make explicit that our experiments were done in vitro every time ADE is mentioned and we rephrased some lines to give a more cautious interpretation of the clinical association of our findings.

Please see below for specific comments.

Comments:

1. Figure 1b, what was the ages within each group (pre-TBE vaccinated and non-vaccinated), this should be included in the graph.

Response: The figure has been modified accordingly. The histogram now includes a separation for TBEV-vaccinated and unvaccinated groups, along with an additional table presenting cohort characteristics classified by TBEV pre-vaccination status. The order of the panels has been rearranged to optimize the use of page space.

2. The TBE vaccination had occurred >4 weeks before YFV vaccination; it would have been interesting to see what exact the time point for the last dose was in the individuals and number of doses.

Response: We agree it would have been very interesting. Unfortunately, for the 250 participants of cohort-1, we do not have this data. Current guidelines for TBEV vaccine administration recommend 3 doses and a boost dose every 5 years. Naturally, not all the TBEV-vaccinated participants might have adhered to this recommendation.

Nevertheless, we do have TBEV-PRNT data and also YF and TBEV-specific IgG antibodies at baseline which are good indicators of the strength of the anti-TBEV immunity at the moment of YF17D vaccination.

Since not all individuals respond to vaccines with equal strength and, together with the variations in the number of vaccine doses and the time intervals between vaccinations, our serology data provides an informative perspective in this context about the strength of anti-TBEV immunity at the time of YF17D vaccination. However, we would have welcomed the opportunity to incorporate this information in our analysis if it had been available.

3. Did the authors compare the viremia in the 2 groups? Did previous TBE vaccination affect the viremia and/or viral clearance in any way?

Response: We agree with the reviewer regarding the importance of measuring viremia in the context of our study. We have made extensive efforts to measure viremia in both fresh and cryopreserved plasma and serum samples. In our efforts, we have used different approaches including commercial kits, such as the Yellow Fever Virus Genesig Advanced Kit (PrimerDesign™, London, UK), as well as methods described in the literature (Drosten et al. 2001, Fischer et al. 2017) including the sensitive one-step RT-qPCR analysis method, which minimizes potential contaminations (Domingo et al., 2012). However despite these efforts were unable to obtain reliable viremia data. We had false positive results in baseline samples and also positive signals in less than half of the donors tested. This challenge is likely attributed to the exceptionally low levels of YF17D replication, resulting in viremia that falls below the sensitivity limit of the assays at the time points we investigated (days 0, 3, and 7).

Chan et al. in Nature Microbiology in 2016 did report an increased and prolonged YFV viremia in JEV pre-vaccinated individuals, but we could not validate this observation in our cohort. We are not alone in having difficulties with measuring viremia of YF17D; for instance, Sandberg et al. (2021) managed to detect viremia in less than 50% of the tested individuals. And Akondy et al (2015) identified the peak of viremia on day 5 pv, a timepoint that we don't have available for this cohort.

4. Figure 2F; surprisingly low levels of plasmablasts, however, in this study frozen PBMCs are used in the plasmablast experiments, in general the recovery of plasmablasts in frozen samples are significantly lower and fresh PBMCs should be used for these experiments. In the literature, data show a plasmablast expansion in YFV vaccinated naïve donors peaking around 10-14 days (Wec et al, Kohler et al). Thus these data are in conflict with the literature and might be explained by methodological problems. Flowcytometry staining on the B-cell populations would have added to the study

Response: Unfortunately we could not use fresh samples for these experiments. We acknowledge that the use of cryopreserved samples may have reduced our capacity to detect plasmablasts in the ELISpot assay, however we were glad to observe the high number of YF-specific B cells we could detect upon the differentiation of memory B cells into antibody-secreting cells (Figure 2g and 5e).

To measure YF-specific IgG-secreting plasmablast, we used cryopreserved samples without further differentiation and the number of plasmablast in the TBEV-unvaccinated group was very low, but it contrasted clearly with the high number of spots we counted for TBEV-pre-vaccinated individuals.

We believe we are not in conflict with previous literature for the following reasons:

- 1) as the reviewer noticed, we had to use cryopreserved samples.
- 2) Sandberg et al J Immunol 2021, measured YF-specific plasmablast after YF-vaccination in a Fluorospot assay and, at day 14, for cryopreserved samples, they detected for most participants < 50 spots in 10⁶ PBMC. Given to sample availability reasons, we could only plate 100.000 PBMC per well (which already cover the full surface), assuming the same plasmablast frequency, 100.000 PBMC would have resulted in <5 spots per well, which is approximately what we could count for TBEV-unvaccinated individuals.
- 3) We are quantifying IgG-secreting cells, and a big proportion of the YF-specific plasmablast may secrete IgM antibodies (Quách et al J Immunol. 2016). Besides, Kohler et al, Wec et al and also

Sandberg et al, found the plasmablast response to YF17D vaccination to peak on day 14 in a frequency of approximately 5% of B cells. Having plated 100.000 PBMC, of which 5% may be B cells of which 5% may be plasmablasts and only a fraction are IgG-secreting and antigen-specific, probably we were in the borderline of the detection limit for the TBEV-unvaccinated group. It is mostly for sample availability reasons that we could not plate more cells and we prioritized cells for differentiation with R848 and IL2 into ASC.

All considered, despite the low detection of plasmablasts in TBEV-unvaccinated, due to the notable difference we observed with the TBEV-vaccinated group, we considered this result to be valuable and reliable. As a result, it has been included in Figure 2.

The reviewer commented that flow-cytometric identification of B cell populations would have added to this study. We have recently generated flow cytometry data on the B and T cell response to YF-17D vaccination. This analysis is still ongoing and our observations are preliminar. We also wish to use this dataset on a separate manuscript with a different scope to the manuscript here reviewed.

Altogether, we acknowledge the technical limitations of the ELISpot assay. Nevertheless, the comparison between TBEV pre-vaccinated groups showed clear significant differences that we consider valuable, informative, and worth showing in Figure 2 of this manuscript.

5. As the main mediator of neutralization on day 28 was the IgM fraction (75%), later time points at 6-12 months would have added important information to the study.

Response. We agree and we now have added additional data from samples collected one-year after vaccination. The IgM fraction is still important and neutralizing (consistent with Gibney KB, et al. *The Am. J. Trop. Med. Hyg.*, 2012). Please find these results in Figure 7 and further discussed in the discussion section.

6. Figure 3 A, also a plot with sera from a YFV vaccinated TBE-neg donor should be shown. It would also be of interest to see how the ADE of YFV vaccinated donors would look at sera from later time points.

We have added a representative plot from a TBEV-negative donor baseline sample.

The YF-ADE after YF17D vaccination results in neutralization and enhancement of the homologous virus infection only occurs at subneutralizing antibody concentrations. We did not measure this parameter on day 28 samples for the study samples.

Regarding DENV ADE with sera from later timepoints, we have added in Figure 7G the results with one-year post-vaccination samples. TBEV-pre-vaccinated expanded antibodies mediating DENV-ADE one year pv whereas TBEV-unvaccinated did not show enhancing capacity at that timepoint either.

7. Figure 4. As mentioned under point 2. The time since the last dose of TBE vaccination would have been of major interest. Also, would this cross reactivity also be valid for other inactivated flavivirus vaccinees (only baseline in this context)? E.g. JEV

Response: We do not have the cohort to address this question, we could only speculate based on what others have observed. Nevertheless, eliciting a cross-reactive response seems inevitable after natural infections and vaccinations with other flaviviruses, especially for inactivated “killed” vaccines like JEV vaccine.

Chan et al, 2020 found that cross-reactive antibodies induced by JEV vaccine could cross-react with YF17D leading to enhanced immunogenicity. They did not look at the antibodies elicited after YF17D in these individuals but we speculate they would share similarities with our cohort.

Saito *BMC infectious diseases* 2016 showed that JEV vaccinees serum could enhance DENV infection. Lai et al *Jvirol* 2008 also found that natural dengue infection mostly generated cross-reactive antibodies against DII-FL. A very interesting study by Malafa et al *PLOS Negl. Trop. Dis.* 2020,

described that natural ZIKV infection was also expanding cross-reactive antibodies in individuals previously vaccinated with TBEV, YF17D or both.

In summary, although we could only describe with certainty what happens in the subsequent vaccination of TBEV with YF17D, we believe that other vaccines or natural infections before YF17D exposure would lead to the same outcome as we have described in our cohort.

8. Would the cross-reactive antibodies decrease as the antibodies affinity mature and the IgG antibodies also wane in serum over time? A later time point preferentially after one year would certainly add to the story. It would also be important to perform the in vitro infectious experiments at a later time point to understand if the in vitro data on ADE on heterologous flaviviruses still stand.

Given the relevance of this question, we have performed additional experiments on a separate cohort that we described above. The reviewer may find the results in Figure 7 and described and discussed in the text. One year post-vaccination, TBEV-pre-vaccinated individuals enhanced DENV infection, and TBEV-unvaccinated individuals remained unable to enhance DENV infection via ADE- The further B cell development did not result in increased cross-reactivity in the TBEV-unvaccinated group. We did not observe a decay in the enhancing capacity (although, since samples belong to different cohorts, we cannot match the observations longitudinally). Basically, one year pv samples recapitulated the enhancing capacity that we observed already by day 28 on TBEV-pre-vaccinated individuals.

9. Page 11, line 344 in vitro should be included after infection.

It has been added and also every time at which ADE of DENV and ZIKV infection was mentioned

10. Discussion page 14, line 428. The authors should add that the enhanced infection is observed in vitro.

It has been clarified

Reviewer #3 (Remarks to the Author):

In this study, the authors examined the impact of prior vaccination against tick-borne encephalitis virus on the immune response induced by the yellow fever virus vaccine YF17D. Their findings, presented in a very well-written manuscript and supported by a compelling array of assays, show that the IgG response is skewed toward the pan-flavivirus E-Fusion Loop epitope.

Flaviviruses are antigenically related, and during natural infection or vaccination, they generate an antibody response that can cross-react with flaviviruses from the same serocomplex as well as from other serocomplexes, as shown in this study. Understanding the complex immune response due to sequential infection is important to assess the risk (symptomatic or severe infection) during subsequent infection and adapt vaccine strategies.

Comments :

1- If the word count allows, the authors should change the title (Prior flavivirus immunity by Prior TBEV immunity) in order to align it with the scope of the study.

We would like to stay with the title we propose, with minor changes. Nevertheless, we have rephrased the abstract, discussion and concluding remarks throughout the text to make it clear we refer to prior TBEV immunity and not other flavivirus.

2- Rephrase or omit the last sentence of the abstract. Together, these results suggest that with prior *TBEV vaccination* exposure *to* YF17D *leads to an antibody response skewed toward the FL epitope which can potentially increase the risk of severeHere the authors do not measure how an increase in anti-FL IgG correlates with the severity risk of subsequent DENV or ZIKV

infection. They only show that anti-FL antibodies mediate ADE in vitro. As a result I would suggest mentioning the potential risk without making any assumptions

The reviewer is right and we have rephrased the abstract

3- Supplementary tables do not have a number/title

This has been corrected

4- The authors should include in the results section a more detailed description of the cohort. This should encompass information about the geographical area where the cohort was conducted, a clarification of the inclusion and exclusion criteria applied to the cohort, details regarding the periods of recruitment, and specific dates of data collection and give some more details in the methods section.

We have now retrospectively registered the cohort in a public repository ISRCTN registry (ISRCTN17974967). There, more detailed information about the cohort can be found including inclusion and exclusion criteria.

In the results section, we have now included a summarizing table in figure 1 with essential information (age, sex and BMI) and we have added in the supplementary table 1 information on participants' ethnicity and the year of recruitment.

5- a) The demographics and clinical backgrounds of both groups (TBEV vaccinated vs. non-vaccinated) should be included in Table 1 (the unstratified one), as these represent the two primary groups of interest in the study. It is critical to assess whether there were any significant differences in relevant factors between these two groups. Additionally, if the groups were matched, it is essential to specify the criteria used for matching groups.

We have improved the tables to give the information separated by TBEV groups (Fig.1 d and Supplementary Table 1). See answer to your comment 4.

We did not match the groups. For the study sample collection, TBEV pre-vaccination was not an inclusion or exclusion criterion. This resulted in an imbalance in number of individuals: 139 TBEV pre-vaccinated and 56 naïve. Nevertheless, both subgroups have similar age and sex distribution (see Figure 1c).

b) Since "fever" is mentioned in the participant clinical data, it would be beneficial to indicate whether participants experienced fever in the 4 weeks preceding YF17D vaccination.

The temperature was measured before vaccination (all participants had a normal body temperature) and none of the participants experienced fever after vaccination. We have added a comment about that now in the methods section.

Related to this comment, participants were asked about infections in the 2 weeks before vaccination. Only a small fraction of the cohort underwent an infectious disease 2 weeks before vaccination (n = 45) among those, 3 had urinary infections, 2 gastrointestinal disease, and 40 flu-like symptoms. Having had infection did not associate with the outcome to YF17D vaccine neither did CRP levels at day 0 prior to vaccination.

We have included the information about infections in the last 14 days prior to vaccination in supplementary table 1 and removed information on how often the participants in general had fever episodes per year as we do not think this is relevant in this context.

c) I also recommend that the authors streamline the list of clinical information to include only factors that are directly relevant to the study. For instance, it may not be necessary to include information about siblings or children unless there is a clear connection established between these factors and the antibody response after YF17D vaccination

The reviewer is right and the information in the table has been reduced accordingly.

6- #MAJOR Comment: Please provide a description of the assessment of prior flavivirus immunity (other than TBEV). Include details about the specific flaviviruses tested, the methods employed, and the antigens used.

Response. Our exclusion criteria included individuals who had received a JEV vaccination or had a known infection with a flavivirus. While we couldn't completely rule out the possibility of a past undiagnosed infection with another flavivirus, it's important to note that our cohort was established in Germany, where no common flaviviruses other than TBEV are known to circulate. Additionally, out of the 250 individuals recruited in our cohort, only 24 had traveled to endemic areas (e.g., Brazil, Thailand, etc.). Therefore, the likelihood of exposure to other flaviviruses is very low.

We did not conduct specific tests for other flaviviruses except for the 41 individuals who were tested at baseline samples with the IIFT. This test revealed the expected high baseline cross-reactivity across the nine flaviviruses tested in TBEV-pre-vaccinated individuals, while TBEV-unvaccinated individuals showed no cross-reactivity.

Please see in addition our response to point 4 and 5 of reviewer 1 above.

In summary, we are confident that the TBEV-unvaccinated group truly represents flavivirus-naïve individuals. The TBEV-vaccinated group has been exposed to TBEV but we cannot rule out the (unlikely) possibility of another flavivirus. In the unknown group, there is room for cases that perhaps did not neutralize TBEV but had specific IgGs but they were excluded from analysis.

7- Define the term "baseline." While I assume it refers to the time point immediately before YF17D vaccination, it's important to explicitly specify this definition as it may not be inferred by the reader. The reviewer is right and we have defined that with baseline we mean the time point immediately before YF17D vaccination.

8- #MAJOR Comment: It is crucial for the authors to include p-values in each panel of the figure to provide a clear indication of the significance of the observations
This has been amended in figures and figure captions.

9- Figure 2C "The IgM titer reached a plateau between day 14 and 28 pv and was comparable in both groups of vaccinees" What was the test used for comparing groups before IgG depletion? It looks like Anti-YF17D IgM titers are lower in the TBEV vaccinated vs TBEV un-vaccinated groups without depletion, particularly on days 14 and 28. Was this difference significant?

The reviewer is right, in undepleted serum, the quantified IgM titer in TBEV-unvaccinated participants is significantly higher than for TBEV-pre-vaccinated. However, After depleting IgGs specifically using the Protein G spin plates (Thermo Fisher, 45204), the remaining IgM titers were not statistically different between TBEV groups. This is why we chose not to emphasize the significance in the undepleted serum, as we believe it is a result of competition for binding to the antigen with the IgG fraction (which is much higher in TBEV-vaccinated individuals).

In Figure 2C and D we observed how both subgroups IgM response peaked at the same timepoint and shared a similar kinetics.

We have added the p values in this figure

10 - #MAJOR Comment: I strongly recommend that the authors include a supplementary figure displaying the characterization and quality control data for the set of recombinant proteins and domains used in the study. While I believe that the constructs behave as described it is important to provide empirical data to support it. Specifically, it would be valuable to show the binding of a range of monoclonal antibodies, including multiple FL antibodies targeting slightly different epitopes than 4G2, EDE1 and 2

We have now added an additional supplementary figure 6 with more data that includes a characterization of the recombinant proteins used in this study.

- Please, see our response to point 8 of reviewer 1, where we provide a detailed explanation of the results presented in the new supplementary figure and elaborate on the suitability of the recombinant proteins used in this study based on results presented here and previous publications from our lab and others.
- Regarding the reviewer's point on showing the binding of a range of monoclonal antibodies, including multiple FL antibodies targeting slightly different epitopes than 4G2, EDE1, and 2: Unfortunately, we only possess the 5A and 4G2 antibodies with precisely mapped epitopes and E21.3 an in house produced antibody binding a non-defined epitope in DII outside of the fusion loop epitope (see extended figure 6c) EDE1 and 2 have not been mapped for yellow fever. Drawing parallels with DENV, EDE-2-like antibodies cannot exist for YFV since the E protein is not glycosylated and there are no reported EDE-like antibodies for YFV. As far as our knowledge extends, only four studies have produced human antibodies against YFV: Daffis et al. (2005) identified mAbs with potent neutralizing capacity elicited following wild-type YFV infection and vaccination, including the 5A mAb we utilized. Li et al. Innovations, 2022, Doyle et al. mBio 2022, and Wec et al. PNAS, 2020 are the other three studies. Wec et al. produced a large panel of mAbs using sE to sort antigen-specific MBC, but their epitope was not finely defined. Moreover, since they used monomeric sE to sort B cells, the specificities they identified are monomeric. In our study, we propose the breathing-dimer as a valid construct to identify and map quaternary antibodies against YFV. We have demonstrated the presence and relevance for neutralization of EDE-like antibodies elicited by the YF17D vaccine. Subsequent studies can leverage the breathing-dimer strategy to pioneer in the identification and mapping of EDE antibodies in YFV.

Altogether, the protein preparation was performed according to validated procedures that our lab use routinely for structural studies, and our biochemical characterization show that the constructs have the expected characteristics of size and binding. Even though we have not solved the structures of these constructs, we are confident of their suitability for this study.

11- line 421 – clarify : “In addition, we show that higher enhancing titers* are associated with stronger neutralizing antibody response...”

***titers prior to YF17D vaccine yes ?**

That is correct, we have clarified this point

12- line 435 “the predisposition” – high magnitude of anti-FL IgG titers is not a causal factor. FL-Abs are likely not enough to predispose someone to severe dengue. Please rephrase the sentence

We have restructured the discussion and, specifically, we have rewritten that paragraph to avoid overinterpretation of our results.

Reviewers' Comments:

Reviewer #1:

Remarks to the Author:

I would like to thank the authors for a very considered and careful revision of their manuscript. The findings are very interesting and the interpretation is now well balanced.

Reviewer #2:

Remarks to the Author:

The authors have adequately addressed the comments in the revised manuscript and rebuttal letter.

Reviewer #3:

Remarks to the Author:

The authors have successfully addressed all the concerns that I raised and have improved the clarity of the manuscript. I do not have any further comments.